# Unified Deployment-Aware Evaluation of Open Reasoning Language Models

## Abstract

Open reasoning language models are often compared using mixed sample sizes, partially standardized prompts, and accuracy-centered summaries, making practical model selection difficult to interpret. We present a unified evaluation of seven open reasoning language model configurations across ARC-Challenge, GSM8K, MATH levels 1–3, and TruthfulQA MC1 under zero-shot, chain-of-thought (CoT), and few-shot CoT prompting. The study uses a complete matched-core matrix with 238 examples per model–dataset–strategy condition and a larger-sample robustness matrix with 500 examples for ARC-Challenge, GSM8K, and TruthfulQA MC1. MATH L1–L3 uses all 238 Level 1–3 examples in the configured MATH test split. In addition to accuracy, we report Wilson confidence intervals, latency, peak video random access memory (VRAM), weighted aggregate performance, Pareto-efficient operating points, prompt-sensitivity metrics, parser-aware diagnostics, and load-mode audits. Under the corrected principal evaluation, which replaces the original-order TruthfulQA MC1 rows with deterministic option-count-aware shuffled-choice results, Gemma-4-26B-A4B with zero-shot prompting is the strongest fixed weighted operating point, reaching 0.776 in the matched core and 0.765 in the larger-sample matrix. Gemma-4-E4B with few-shot CoT remains a close lower-memory operating point, reaching 0.758 and 0.749, respectively. Repeated-subset analysis shows that the leading configurations remain sufficiently close for deployment tradeoffs to remain important. The corrected TruthfulQA MC1 audit further shows that the original prepared representation placed the correct answer in position A for all 500 selected examples. We therefore use deterministic option-count-aware choice shuffling, gold-label remapping, a prompt without the previous A–E restriction, an A–Z extractor, balanced few-shot demonstrations, and a common 256-token generation cap. Under this corrected 500-example protocol, the always-A baseline is 0.238, and Gemma-4-26B-A4B with few-shot CoT is the strongest TruthfulQA condition with accuracy 0.698. We further find that prompting changes ranking order rather than shifting all models uniformly. Compatibility diagnostics also show that some apparent failures, especially for Phi-4-Reasoning on GSM8K and MATH-style tasks, reflect deployment-relevant robustness and interface-adherence problems under the shared evaluation pipeline. These results support framing open-model evaluation as a deployment-aware, multi-objective operating-point problem rather than as a single-score leaderboard exercise.

## 1 Introduction

Open large language models (LLMs) are increasingly evaluated not only by raw benchmark accuracy, but also by robustness, efficiency, and deployment cost. Recent evaluation frameworks and surveys have argued that reliable LLM assessment requires standardized protocols, broader metric coverage, and explicit attention to inference efficiency rather than accuracy alone Liang et al. (2023); Laskar et al. (2024); Chang et al. (2024). This is especially important for reasoning-oriented models, whose observed performance can change substantially under different prompting strategies such as zero-shot prompting, chain-of-thought (CoT) prompting, and few-shot CoT prompting Wei et al. (2022); Kojima et al. (2022); Wang et al. (2023); Sclar et al. (2024).

At the same time, recent work has highlighted that evaluation conclusions can be sensitive to seemingly minor methodological choices. Prompt formatting alone can produce large performance swings in open models, which raises concerns about comparing models under arbitrarily chosen prompt templates Sclar et al. (2024). More broadly, recent reviews of LLM evaluation have emphasized that inconsistent dataset choices, heterogeneous protocols, and incomplete reporting often make cross-paper conclusions difficult to compare or reproduce Laskar et al. (2024); Chang et al. (2024). These concerns are particularly relevant in reasoning evaluation, where benchmark results are often used to support strong claims about model capability.

This paper argues that the problem is not only incomplete standardization, but also incomplete framing. In many practical settings, model choice is not a pure leaderboard problem. A configuration with the highest aggregate score may be too slow, too memory-intensive, too prompt-sensitive, or too task-specific to be the most attractive deployment choice. Conversely, a configuration that is not the absolute score leader may offer a substantially better operating point once latency, video random access memory (VRAM), prompting stability, and task-dependent behavior are considered together. Our aim is therefore not merely to compare several open reasoning models, but to show that open-model evaluation should move from single-score ranking toward deployment-aware operating-point analysis.

We study that problem directly through a unified evaluation of seven open reasoning language model configurations on four widely used benchmark tasks: ARC-Challenge (Clark et al., 2018), GSM8K (Cobbe et al., 2021), MATH levels 1–3 (Hendrycks et al., 2021), and TruthfulQA MC1 (Lin et al., 2022). Each model is tested under three prompting strategies: zero-shot, CoT, and few-shot CoT.

The evaluation has two complementary layers. The first is a complete matched core matrix in which, within each dataset, the same 238 selected examples are used across all model–strategy configurations. The second is a larger-sample robustness matrix in which ARC-Challenge (Clark et al., 2018), GSM8K (Cobbe et al., 2021), and TruthfulQA MC1 (Lin et al., 2022) use 500 examples per condition, while MATH L1–L3 uses the complete level-1–3 portion of the configured MATH test split, which contains 238 examples (Hendrycks et al., 2021). This design preserves a fully matched cross-benchmark comparison while also testing whether the main deployment-aware conclusions persist under larger non-MATH samples.

The purpose of the study is not to argue that one current model family is universally best. Instead, the goal is to understand what becomes visible when open reasoning models are evaluated under a fully unified and deployment-aware protocol.

We focus on the following research questions:

1. How do rankings change when model–dataset–strategy conditions are evaluated under a shared matched-size protocol?

2. Do the main operating-point conclusions persist when ARC-Challenge, GSM8K, and corrected deterministic option-count-aware TruthfulQA MC1 are evaluated at 500 examples per condition, while MATH L1–L3 uses the complete 238-example level-1–3 test subset?

3. Does the highest weighted-score configuration coincide with the most attractive practical operating point under latency and memory constraints?

4. How much of the measured prompt sensitivity on multiple-choice tasks reflects answer-only formatting versus rationale-allowed prompting with an explicit final-answer line?

5. Which apparent failures are ordinary wrong answers, and which are better attributed to parser compliance, missing final answers, or model–pipeline interface mismatch?

6. How much headroom is suggested by cross-task complementarity under the corrected matched-core and larger-sample matrices after replacing original-order TruthfulQA MC1 rows with corrected deterministic option-count-aware shuffled-choice results?

The main contributions of the paper are as follows.

1. We present a two-layer unified benchmark protocol: a complete matched 238-example core matrix across all seven models, four benchmarks, and three prompting strategies, together with a larger-sample robustness matrix for ARC-Challenge, GSM8K, and corrected deterministic option-count-aware TruthfulQA MC1 at 500 examples per condition, while MATH L1–L3 uses the complete 238-example level-1–3 test subset.

2. We distinguish descriptive weighted-score leadership from practical deployment attractiveness, and we formalize that distinction through Pareto-frontier analysis, deployment-budget summaries, resource-normalized efficiency metrics, and weight-sensitivity analysis.

3. We quantify prompt sensitivity under the original shared prompt family and through a 500-example ARC-Challenge multiple-choice prompt-protocol ablation that separates answer-only rules from rationale-allowed final-answer prompting, while TruthfulQA MC1 is handled through a corrected deterministic option-count-aware shuffled-choice audit.

4. We quantify cross-task complementarity through an oracle task-aware upper bound after correcting the TruthfulQA component, and we show that the remaining routing headroom is modest relative to the best fixed operating point.

5. We provide parser-aware compatibility diagnostics showing that some apparent benchmark failures, especially for Phi-4-Reasoning under GSM8K and MATH-style settings, are better explained by missing extractable final answers, trace-like outputs, or format mismatch than by ordinary wrong-answer behavior alone.

6. We release code, prompt configurations, selected sample IDs, extraction logic, raw outputs or compact output summaries, load-mode audits, and software-environment details to support reproducibility under the reported bf16 H100 batch-size-one setting.

Taken together, these contributions support a central claim: open reasoning model evaluation should be treated as a multi-objective operating-point selection problem rather than as a single-score leaderboard exercise.

The rest of the paper is organized as follows. Section 2 reviews prior work on reasoning benchmarks, prompting-based reasoning, prompt sensitivity, and holistic LLM evaluation. Section 3 describes the unified methodology in detail. Section 4 reports the main empirical results, including deployment-aware operating points, oracle routing headroom, and compatibility diagnostics. Section 5 discusses what these results imply for model evaluation and deployment, and also presents limitations and future work. Section 6 concludes the paper.

## 2 Related Work

### 2.1 Reasoning Benchmarks for Large Language Models

A substantial body of recent work evaluates LLMs on benchmarks designed to probe reasoning rather than shallow pattern matching. ARC-Challenge was introduced as a science question answering benchmark specifically constructed to be difficult for retrieval and co-occurrence baselines, thereby emphasizing more substantive reasoning ability Clark et al. (2018). GSM8K has become a standard benchmark for grade-school mathematical reasoning and multi-step arithmetic problem solving Cobbe et al. (2021). MATH extends this line of evaluation to competition-level mathematics and was designed to test mathematical problem-solving ability at a substantially higher level of difficulty Hendrycks et al. (2021). TruthfulQA provides a complementary perspective by measuring whether models reproduce common human falsehoods and misconceptions, rather than merely maximizing answer plausibility Lin et al. (2022). Taken together, these datasets cover distinct but complementary aspects of reasoning, factual reliability, and answer validity.

## 2.2 Prompting-Based Reasoning and Prompt Sensitivity

Prompting strategy has become central to reasoning evaluation. Chain-of-thought prompting showed that providing intermediate reasoning demonstrations can substantially improve model performance on arithmetic, commonsense, and symbolic reasoning tasks Wei et al. (2022). Zero-shot CoT later demonstrated that explicit reasoning can also be elicited without exemplars through simple reasoning-trigger phrases, suggesting that prompt design itself is a major determinant of observed capability Kojima et al. (2022). Self-consistency further showed that decoding strategy interacts with CoT prompting in important ways, yielding additional gains by aggregating across multiple sampled reasoning paths Wang et al. (2023).

More recent work has shown that prompt sensitivity is not merely a nuisance variable but a serious methodological issue. Sclar et al. show that open LLMs can be highly sensitive to meaning-preserving prompt formatting changes in few-shot settings, and argue that reporting a single prompt format can mischaracterize model quality Sclar et al. (2024). Related robustness work has similarly found that LLMs are vulnerable to prompt perturbations and adversarial prompt variations, reinforcing the need to treat prompting choices as an experimental factor rather than a minor implementation detail Zhu et al. (2023); Gan & Mori (2023).

## 2.3 Holistic and Standardized LLM Evaluation

A parallel line of work argues that LLM evaluation should be broader, denser, and more standardized. HELM is especially influential in this regard, because it explicitly promotes scenario coverage, multi-metric reporting, and standardized comparison across models Liang et al. (2023). Recent survey papers further document that heterogeneous evaluation setups, incomplete reporting, and protocol mismatch remain widespread problems in the literature Chang et al. (2024); Laskar et al. (2024). In the open-model setting, recent leaderboard efforts have also attempted to improve comparability and reduce evaluation artifacts. For example, the Open-LLM-Leaderboard paper argues that multiple-choice evaluation can hide issues such as selection bias and random guessing, and proposes open-style evaluation to better reflect model capability Myrzakhan et al. (2024). Tooling work such as LLMBox likewise emphasizes unified interfaces for training, inference, and evaluation, reflecting the field's broader shift toward reproducible and systematized benchmarking Tang et al. (2024).

## 2.4 Position of the Present Study

The present study is closest in spirit to holistic and standardized benchmarking, but it makes a narrower methodological intervention. Rather than proposing a new benchmark, we examine what can be learned from evaluating existing open reasoning models under a unified protocol with matched sample sizes, fixed prompting families, and deployment-oriented measurements.

The contribution is not a larger leaderboard, but a deployment-aware evaluation pattern that combines a complete model–dataset–prompt matrix, shared prompt construction and extraction rules, paired resource measurements, prompt-sensitivity analysis, parser-compliance diagnostics, and operating-point interpretation. Compared with leaderboard-style evaluations that primarily report task accuracy, the present analysis asks whether the same configuration remains attractive when memory, latency, prompt protocol, extraction compliance, and task composition are considered jointly.

The broader claim is methodological rather than universal. The paper does not claim that the exact ranking of the evaluated models will transfer unchanged to other releases, hardware, batch sizes, precision modes, prompts, or extraction systems. Instead, it shows how a deployment-aware protocol can reveal tradeoffs that a single-score accuracy ranking can hide.

The protocol is also intended to be extensible to new models, benchmarks, prompting strategies, and deployment settings. Extending it to another model requires recording the model identifier, revision evidence, tokenizer and chat-template behavior, generation configuration, and load-state audit. Extending it to another benchmark or prompting strategy requires a standardized example schema, selected example identifiers, a versioned prompt definition, a task-specific strict extraction rule, and new deployment measurements in the target hardware and software environment.

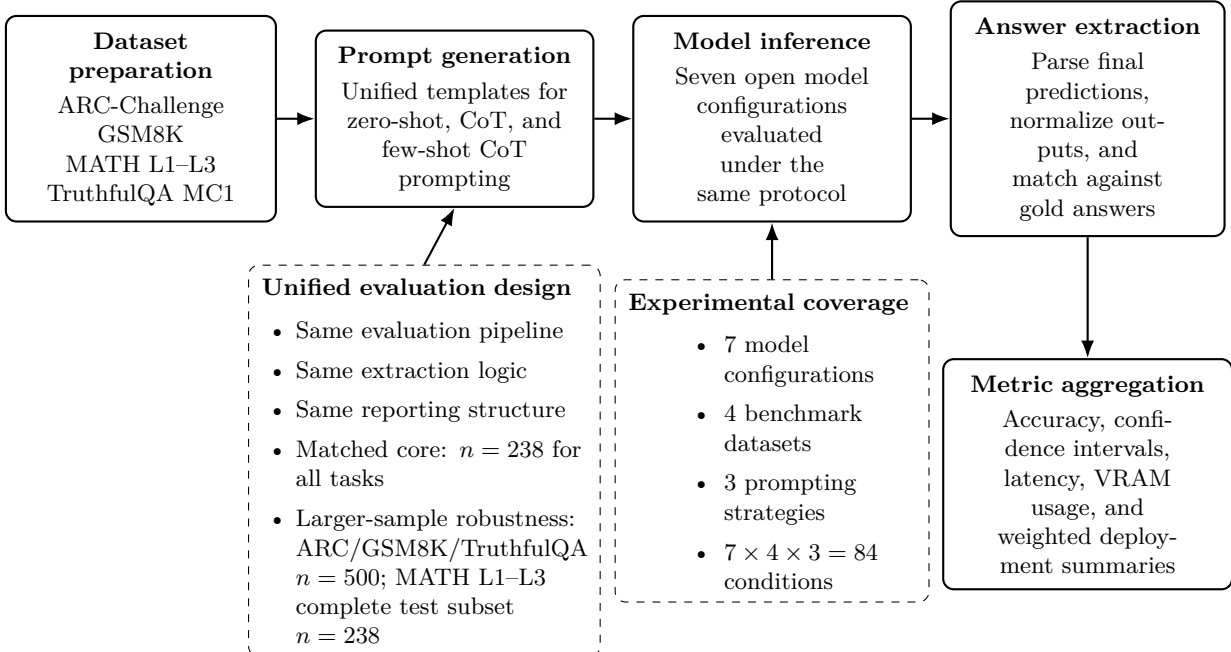

Figure 1: Overview of the unified evaluation workflow. The study uses a single standardized pipeline for dataset preparation, prompt construction, model inference, output extraction, and metric aggregation across all model–dataset–strategy conditions. The matched core matrix evaluates seven open model configurations on four benchmarks under three prompting strategies, yielding 84 directly comparable conditions at 238 examples per condition. A larger-sample robustness matrix evaluates ARC-Challenge, GSM8K, and TruthfulQA MC1 at 500 examples per condition, while MATH L1–L3 uses the complete level-1–3 portion of the configured MATH test split, which contains 238 examples.

## 3 Methodology

We evaluate seven open reasoning language model configurations across four datasets and three prompting strategies under a fully unified protocol. Figure 1 illustrates the workflow. The pipeline contains five main stages: dataset preparation, prompt generation, model inference, answer extraction, and metric aggregation.

### 3.1 Benchmark tasks

The unified evaluation uses four benchmark datasets: ARC-Challenge for multiple-choice scientific reasoning Clark et al. (2018), GSM8K for grade-school mathematical reasoning Cobbe et al. (2021), MATH for mathematical problem solving Hendrycks et al. (2021), and TruthfulQA MC1 for multiple-choice truthfulness evaluation Lin et al. (2022). These benchmarks were chosen to cover complementary evaluation dimensions: ARC-Challenge emphasizes science-oriented reasoning under a multiple-choice format, GSM8K emphasizes multi-step grade-school mathematical reasoning, MATH L1–L3 emphasizes more advanced mathematical problem solving, and TruthfulQA MC1 emphasizes truthfulness and resistance to plausible but misleading answers.

The benchmark preparation has two complementary sample-size views. The matched core view uses 238 examples per model–dataset–strategy condition for all four benchmarks. This produces a complete and directly comparable seven-by-four-by-three matrix with 84 conditions and 19,992 evaluated examples. The larger-sample view uses 500 examples per condition for ARC-Challenge, GSM8K, and TruthfulQA MC1, while MATH L1–L3 uses the complete level-1–3 portion of the configured MATH test split, which contains 238 examples.

We use the phrase "MATH L1–L3" to denote the complete level-1–3 portion of the configured MATH test split used in this study. Under the configured benchmark source, the MATH test split contains 500 examples in total, of which 43 are Level 1, 90 are Level 2, and 105 are Level 3, yielding 238 level-1–3 examples. The evaluation therefore uses all available level-1–3 test examples rather than a hand-curated subset. Expanding this component to 500 examples would require either including Level 4–5 problems or using training-split examples, which would change the benchmark definition. Consequently, the larger-sample robustness matrix should be interpreted as a 500-example robustness check for ARC-Challenge, GSM8K, and TruthfulQA MC1 while retaining the complete MATH L1–L3 test component.

For reproducibility, the selected example identifiers are saved in the corresponding index files and archived with the benchmark code and configuration files. The preparation pipeline applies a seed-controlled shuffle where sampling is required and stores the selected index set before inference. For MATH L1–L3, the level filter yields exactly 238 test examples, so all filtered examples are retained. The matched core and larger-sample matrices can therefore be reconstructed from the released artifacts.

The four benchmarks were selected to span complementary deployment-relevant failure modes rather than to exhaustively represent all reasoning workloads: multiple-choice science reasoning, arithmetic word-problem reasoning, filtered mathematical problem solving, and truthfulness under plausible distractors. This combination allows the analysis to test whether a single model–prompt operating point remains attractive across heterogeneous reasoning formats.

## 3.2 Model configurations

The model set was chosen to cover several practically relevant open-model regimes rather than to form an exhaustive leaderboard: dense and mixture-of-experts architectures, smaller and larger active-parameter footprints, and three prominent open model families with reasoning-oriented releases. This selection supports controlled operating-point analysis across resource scales, but the conclusions are limited to these model families and release interfaces.

The evaluation includes seven open model configurations:

- Gemma-4-26B-A4B

- Gemma-4-E2B

- Gemma-4-E4B

- Phi-4-Mini-Reasoning

- Phi-4-Reasoning

- Qwen3-30B-A3B

- Qwen3-8B

The current project configuration records the following architecture and parameter metadata. Phi-4-Mini-Reasoning is a dense model with 3.8B total parameters and 3.8B active parameters. Gemma-4-E2B is a mixture-of-experts (MoE) model with 5.0B total parameters and 2.0B active parameters. Gemma-4-E4B is an MoE model with 8.0B total parameters and 4.0B active parameters. Qwen3-30B-A3B is an MoE model with 30.0B total parameters and 3.0B active parameters. Gemma-4-26B-A4B is an MoE model with 26.0B total parameters and 3.8B active parameters. Qwen3-8B is a dense model with 8.0B total parameters and 8.0B active parameters. Phi-4-Reasoning is a dense model with 14.0B total parameters and 14.0B active parameters.

Official release references for the evaluated model families include the Gemma 4 release materials for Gemma-4-E2B, Gemma-4-E4B, and Gemma-4-26B-A4B Google (2026b;a), the Phi-4-Reasoning technical report and Phi-4-Mini-Reasoning model documentation Abdin et al. (2025); Microsoft (2025), and the official Qwen3 release materials and model documentation for Qwen3-30B-A3B and Qwen3-8B Qwen Team (2025c;a;b).

### 3.3 Prompting strategies

Each model was evaluated under three prompting strategies: `zero-shot`, chain-of-thought (CoT), and few-shot chain-of-thought (few-shot CoT). This design treats prompting as a first-class experimental factor rather than as a minor implementation detail. We therefore do not assume that one prompting method is uniformly best across all model families and benchmark tasks. Instead, we study whether the relative ordering of models changes across prompt conditions.

The prompt family was intentionally kept minimal and uniform across datasets and model families. This was a methodological choice rather than an attempt to reproduce each benchmark's original default prompt format. Because prompt sensitivity is itself a central concern in both the prior literature and the present study, we use a shared prompt scaffold to reduce benchmark-specific prompt engineering and to isolate how ranking behavior changes under a common evaluation interface. The goal is therefore controlled comparability, not prompt optimization for any single benchmark.

The three prompting strategies were selected because they represent common evaluation choices with different inference costs and output-format risks: direct zero-shot answering, explicit CoT prompting, and few-shot CoT prompting with demonstrations. We do not treat these templates as optimal prompts. They are standardized prompt families for measuring sensitivity under a shared interface.

For both the original-order and corrected shuffled-choice TruthfulQA MC1 evaluations, the CoT and few-shot CoT wrappers retain answer-only output rules. These TruthfulQA conditions should therefore be interpreted as controlled prompt-family variants rather than unconstrained rationale-generation evaluations. The ARC-Challenge prompt-protocol ablation separately evaluates rationale-allowed multiple-choice prompting with an explicit final-answer line.

To separate answer-format compliance from rationale allowance, the analysis also includes a 500-example ARC-Challenge multiple-choice prompt-protocol ablation. In that protocol, models may provide up to four short reasoning steps, but the final answer must appear on a separate final line in the form `#### <letter>`, with no output after that line. TruthfulQA MC1 is excluded from this ablation table because substantive TruthfulQA interpretation uses the corrected deterministic option-count-aware shuffled-choice protocol.

For the corrected TruthfulQA MC1 few-shot CoT runs, the demonstration block was balanced across answer positions A, B, C, and D to avoid inducing a repeated A-position preference in the few-shot context.

The exact prompt construction procedure for zero-shot, CoT, and few-shot CoT evaluation is documented in Appendix A. The corresponding benchmark code, configuration files, and evaluation summaries are publicly archived at `https://anonymous.4open.science/r/UDAE-D371/`.

### 3.4 Inference Protocol and Implementation

All runs were executed through a common benchmark pipeline. For each model–dataset–strategy condition, the system recorded binary correctness, per-example latency, output token count, token throughput, and peak video random access memory (VRAM). Condition-level accuracy was computed as the mean of the binary correctness values, with Wilson confidence intervals reported for all condition-level estimates.

Inference used the shared runner `experiments/run_benchmark_with_fallback.py`. The runner loads prepared records, constructs the prompt, applies the tokenizer chat template when configured, performs generation, extracts the final answer, and grades it against the gold answer. All evaluated model configurations used `use_chat_template: true`, so prompts were passed through the tokenizer chat-template interface consistently across models.

Generation settings were shared across conditions: temperature was `0.0`, `do_sample=false`, batch size was 1, and the global seed was `42`. Maximum generation lengths were 512 new tokens for GSM8K, 1024 for MATH L1–L3, and 256 for ARC-Challenge and TruthfulQA MC1. Prompt construction and answer extraction used shared interfaces, with strategy-specific wrappers, dataset-specific demonstrations and answer-format rules, and task-specific strict extractors. Appendices B and C document the prompt files, extraction procedures, software environments, selected examples, and artifact contents.

The retained main-result artifacts and the follow-up audit runs were produced on the same H100 server but under separate software environments. Exact package versions for the retained main-run environment and the follow-up audit environment are reported in Appendix C. We distinguish these environments rather than treating the follow-up audit as evidence of every package version used in the original runs.

For the corrected TruthfulQA MC1 evaluation, each selected example was assigned a saved deterministic option-count-aware permutation. The answer choices were permuted, the gold label was remapped, and the same shuffled record was used across all models and prompting strategies. This procedure removes the original all-A gold-position artifact while preserving paired comparisons.

The corrected principal matrices combine retained ARC-Challenge, GSM8K, and MATH L1–L3 results from the main-run environment with corrected TruthfulQA MC1 results from the follow-up audit environment. The corrected TruthfulQA runs preserved the model identifiers, deterministic decoding, selected example identifiers, bf16 loading path, prompting strategies, saved shuffled records, and 256-token generation cap. The remaining software-version differences are treated as a limitation of the combined cross-task analysis.

The main model configuration specified bfloat16 execution. A follow-up load-only audit used the same seven model identifiers and bf16 loading path and found no 4-bit or 8-bit loading, no quantization configuration, no `Linear4bit` or bitsandbytes modules, and bfloat16 parameter tensors for every audited model. Table 23 reports the direct audit evidence. Together with the retained load-mode fields and observed VRAM values, this supports interpreting the reported runs as H100, batch-size-one, bf16 executions under the reported software stacks, rather than as quantized or hardware-invariant measurements.

Latency and VRAM should therefore be interpreted as environment-dependent deployment measurements. Their absolute values depend on the GPU, precision mode, batch size, software stack, kernels, and serving configuration, but remain informative for relative comparison under the shared evaluation setting.

The benchmark code, configurations, selected-example records, extraction logic, and evaluation summaries are available at `https://anonymous.4open.science/r/UDAE-D371/`.

### 3.5 Evaluation metrics

We report the following metrics.

**Accuracy** is the fraction of correctly answered examples for a condition. **Wilson confidence interval** provides uncertainty bounds around the accuracy estimate. **Mean latency** is the average inference time in seconds for a condition. **Peak VRAM** is the maximum observed memory usage in gigabytes for the condition. **Mean tokens per second** is the observed output throughput.

In addition to per-condition results, we compute a weighted accuracy summary using the following task weights: GSM8K 0.40, MATH L1–L3 0.30, ARC-Challenge 0.20, and TruthfulQA MC1 0.10.

This weighted score is used as an illustrative deployment-oriented summary, not as a universal utility function or a complete account of model quality. It places greater emphasis on mathematical reasoning while retaining science-oriented multiple-choice reasoning and truthfulness in the aggregate. Because other deployment settings may imply different task priorities, we report sensitivity to alternative weight vectors and avoid treating small weighted-score gaps as decisive without paired statistical support.

The weight-sensitivity analysis includes the stated weighting, equal weighting, and task-heavy variants. Under corrected TruthfulQA integration, Gemma-4-26B-A4B zero-shot remains the winner for the stated, ARC-heavy, GSM8K-heavy, and MATH-heavy schemes, while Gemma-4-E4B few-shot CoT becomes the winner under equal and TruthfulQA-heavy schemes. This reinforces the deployment-aware interpretation: the preferred operating point depends on the task-weight vector, and TruthfulQA-heavy conclusions must be based on corrected deterministic option-count-aware shuffled-choice results rather than original-order protocol rows. The shuffled-choice TruthfulQA audit is reported separately in Table 9.

Appendix F, Table 22 reports the corresponding weight-sensitivity winners for both the matched core and larger-sample matrices.

For bootstrap intervals over weighted scores, examples are resampled at the dataset level and the task-weighted aggregate is recomputed for each model–strategy configuration. For paired permutation tests, the comparison unit is the aligned example-level correctness difference within each dataset, preserving the paired structure induced by evaluating the same selected examples across configurations. When multiple top configurations are compared, Holm-adjusted p-values are reported in addition to raw two-sided permutation p-values. For larger-sample robustness, the 238-example matched core is compared with the 500-example ARC-Challenge, GSM8K, and TruthfulQA MC1 matrices, while MATH L1–L3 remains the complete 238-example level-1–3 test subset. Repeated 238-example subsets are sampled from the larger non-MATH matrices, with the complete MATH L1–L3 test subset fixed across repetitions.

The bootstrap analysis uses 5000 resamples with random seed 42. The paired permutation tests use 20000 sign-flip iterations with the same random seed. The Holm correction family consists of the seven pairwise comparisons reported in Table 18. For repeated-subset analysis, each draw uses the same sampled example indices for every model–strategy configuration within a dataset. The corrected TruthfulQA shuffled-choice permutation is fixed across all models and prompting strategies.

## 4 Results

This section reports the empirical findings from the matched 238-example core matrix and the larger-sample robustness analyses. The matched core provides complete cross-benchmark comparability across all 84 conditions, while the larger-sample matrices test whether the main ARC-Challenge, GSM8K, and TruthfulQA MC1 conclusions persist at 500 examples per condition. We begin with an overview of the complete matched-size result set and the weighted ranking across model–prompt configurations. We then move from leaderboard-style summaries to deployment-aware interpretation by examining benchmark-specific behavior, Pareto-efficient operating points, routing headroom, and compatibility diagnostics. The goal is not only to identify which configuration attains the highest weighted score, but also to determine which configurations remain attractive once latency, memory, prompting sensitivity, and task-specific behavior are considered together.

### 4.1 Overview of the unified result set

The core result set contains all 84 model–dataset–strategy conditions under the same 238-example protocol. This matched matrix is used for fully comparable cross-benchmark weighted scores, Pareto analysis, prompt-sensitivity summaries, routing headroom, and compatibility diagnostics.

The larger-sample result set contains the same 21 model–strategy conditions for ARC-Challenge, GSM8K, and TruthfulQA MC1 at 500 examples per condition, while MATH L1–L3 uses the complete level-1–3 portion of the configured MATH test split, which contains 238 examples. This second view is used as a robustness check for sample-size sensitivity rather than as a replacement for the matched core matrix.

Table 10 provides a compact summary of the most relevant weighted and benchmark-specific results, while the complete matched-core 84-condition matrix is reported in Appendix D. The larger-sample per-condition matrix is reported separately in Appendix E, Table 15. However, MATH L1–L3 uses the same complete 238-example test subset in both analyses; therefore, its per-condition rows are unchanged from Table 14 and are not repeated in the larger-sample table.

Table 1 presents the top weighted configurations across the 21 model–strategy combinations, and Figure 2 visualizes the same ranking. Figures 3a, 3b, and 3c provide the main views of dataset-specific performance across prompting strategies. The later results subsections then extend this overview in three directions: deployment-aware operating points, oracle routing headroom, and compatibility diagnostics. In particular, Section 4.5 formalizes latency–memory–accuracy tradeoffs, Section 4.6 quantifies task-aware routing headroom, and Section 4.7 examines failure patterns that are not visible from accuracy alone.

Table 1 already shows the central tension of the corrected principal analysis. Gemma-4-26B-A4B with zero-shot prompting achieves the highest corrected stated-weight score at 0.776, while Gemma-4-E4B with few-shot CoT remains close at 0.758 and requires substantially less memory. This means that the corrected score

Table 1: Top weighted configurations under the corrected matched 238-example core protocol. Original-order TruthfulQA MC1 rows are replaced by corrected deterministic option-count-aware shuffled-choice TruthfulQA results. Weighted accuracy uses the stated task weights: GSM8K 0.40, MATH L1–L3 0.30, ARC-Challenge 0.20, and TruthfulQA MC1 0.10.

| Model | Strategy | Weighted Acc. | ARC | GSM8K | MATH | TruthfulQA |
|---|---|---|---|---|---|---|
| Gemma-4-26B-A4B | Zero-shot | 0.776 | 0.945 | 0.794 | 0.693 | 0.618 |
| Gemma-4-E4B | Few-shot CoT | 0.758 | 0.891 | 0.752 | 0.693 | 0.714 |
| Gemma-4-E4B | Zero-shot | 0.751 | 0.899 | 0.790 | 0.630 | 0.664 |
| Gemma-4-E4B | CoT | 0.742 | 0.891 | 0.790 | 0.668 | 0.471 |
| Gemma-4-26B-A4B | CoT | 0.724 | 0.891 | 0.782 | 0.643 | 0.403 |
| Gemma-4-26B-A4B | Few-shot CoT | 0.699 | 0.937 | 0.681 | 0.571 | 0.681 |
| Gemma-4-E2B | Few-shot CoT | 0.688 | 0.727 | 0.710 | 0.689 | 0.521 |

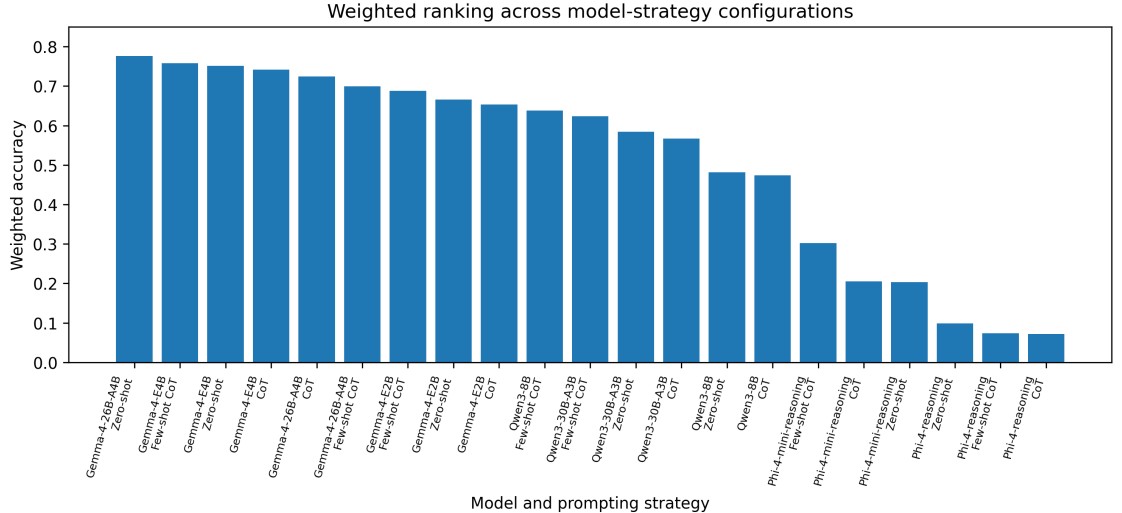

Figure 2: Weighted ranking across model–strategy configurations under the corrected matched 238-example core protocol. Original-order TruthfulQA MC1 rows are replaced by corrected deterministic option-count-aware shuffled-choice TruthfulQA results. Gemma-4-26B-A4B zero-shot is the strongest corrected stated-weight fixed operating point, while Gemma-4-E4B few-shot CoT remains a close lower-memory alternative.

leader and the strongest practical operating point are not automatically the same configuration, even before the more formal deployment-aware analyses are introduced.

## 4.2  Larger-sample robustness

Table 2 summarizes the sample-size structure used for robustness analysis. ARC-Challenge, GSM8K, and TruthfulQA MC1 are evaluated at 500 examples per condition, while MATH L1–L3 uses all 238 level-1–3 examples in the configured MATH test split. All datasets retain complete coverage over the seven models and three prompting strategies.

The larger-sample TruthfulQA MC1 matrix should be interpreted with an additional caveat. The prepared original-order representation placed the correct answer in position A for all 500 selected TruthfulQA examples. Therefore, the original-order TruthfulQA larger-sample rows are retained as protocol and extraction records, but substantive TruthfulQA interpretation relies on the shuffled-choice audit in Table 9.

After replacing the original-order TruthfulQA rows with corrected deterministic option-count-aware shuffled-choice TruthfulQA results, Gemma-4-26B-A4B zero-shot remains the strongest corrected stated-weight

Table 2: Sample-size coverage for the matched core and larger-sample robustness matrices. ARC-Challenge, GSM8K, and TruthfulQA MC1 use 500 examples in the larger-sample matrix. MATH L1–L3 uses the complete level-1–3 subset of the configured MATH test split, which contains 238 examples.

| Dataset | Matched core $n$ | Larger-sample $n$ | Conditions complete |
|---|---|---|---|
| ARC-Challenge | 238 | 500 | 21/21 |
| GSM8K | 238 | 500 | 21/21 |
| MATH L1–L3 | 238 | 238 (complete L1–L3 test subset) | 21/21 |
| TruthfulQA MC1 | 238 | 500 | 21/21 |

Table 3: Top configurations in the corrected matched-core matrix, corrected larger-sample robustness matrix, and corrected repeated-subset analysis under the stated weighting. Original-order TruthfulQA MC1 rows are not used in these principal weighted scores; they are replaced by deterministic option-count-aware shuffled-choice TruthfulQA results with a common 256-token generation cap.

| Model | Strategy | Matched core | Larger-sample | Repeated-subset mean |
|---|---|---|---|---|
| Gemma-4-26B-A4B | Zero-shot | 0.776 | 0.765 | 0.765 |
| Gemma-4-E4B | Few-shot CoT | 0.758 | 0.749 | 0.749 |
| Gemma-4-E4B | Zero-shot | 0.751 | 0.739 | 0.742 |
| Gemma-4-E4B | CoT | 0.742 | 0.732 | 0.734 |
| Gemma-4-26B-A4B | CoT | 0.724 | 0.708 | 0.710 |
| Gemma-4-E2B | Few-shot CoT | 0.688 | 0.693 | 0.698 |
| Gemma-4-26B-A4B | Few-shot CoT | 0.699 | 0.697 | 0.698 |

fixed configuration, reaching 0.776 in the matched core and 0.765 in the larger-sample matrix. Gemma-4-E4B few-shot CoT remains close behind, with corrected scores of 0.758 and 0.749, while requiring substantially less memory. Thus, the corrected analysis preserves the broader deployment conclusion that score, memory, latency, and task emphasis must be considered jointly.

The larger-sample and repeated-subset analyses are not used to claim that the exact third-decimal ordering is fixed. Instead, they support the more limited conclusion that the score leader, the high-performing lower-memory operating point, and the importance of deployment tradeoffs remain visible when ARC-Challenge, GSM8K, and TruthfulQA MC1 are evaluated at 500 examples per condition and when repeated matched-size subsets are drawn from those larger matrices.

To test whether the larger-sample conclusion depends on a particular 238-example draw, we sampled 1000 repeated matched-size subsets. In each iteration, 238 examples were sampled without replacement independently from the 500-example ARC-Challenge, GSM8K, and TruthfulQA MC1 matrices. Because the configured MATH L1–L3 test pool itself contains 238 examples, the MATH component was fixed across repeated subsets. The stated weighted score was then recomputed for every model–strategy configuration.

Table 3 summarizes the corrected matched-core, larger-sample, and repeated-subset weighted rankings under the stated task-weight vector.

Across these 1000 repeated subsets, Gemma-4-26B-A4B zero-shot is the overall weighted-score leader in 926/1000 sampled subsets. Gemma-4-E4B few-shot CoT wins 72/1000 subsets, and Gemma-4-E4B zero-shot wins 2/1000 subsets. The mean rank correlation between the repeated-subset weighted rankings and the matched-core ranking is 0.985 with standard deviation 0.004.

This repeated-subset analysis measures ranking stability under matched-size resampling, whereas the Holm-adjusted paired permutation tests measure pairwise significance among selected top configurations. Thus, the 926/1000 repeated-subset winner frequency in Table 4 and the non-decisive Holm-adjusted top-row comparisons in Table 18 answer different statistical questions rather than contradicting one another.

Table 4: Repeated 238-example subset stability analysis over 1000 sampled subsets from the larger-sample matrices. ARC-Challenge, GSM8K, and corrected deterministic shuffled-choice TruthfulQA MC1 are resampled from their 500-example matrices, while MATH L1–L3 remains fixed because its complete configured level-1–3 test subset contains 238 examples.

| Model | Strategy | Mean | SD | 2.5th | 97.5th | Winner freq. |
|-------|----------|------|-----|-------|--------|--------------|
| Gemma-4-26B-A4B | Zero-shot | 0.765 | 0.008 | 0.750 | 0.781 | 926/1000 |
| Gemma-4-E4B | Few-shot CoT | 0.749 | 0.009 | 0.732 | 0.767 | 72/1000 |
| Gemma-4-E4B | Zero-shot | 0.742 | 0.009 | 0.725 | 0.758 | 2/1000 |
| Gemma-4-E4B | CoT | 0.734 | 0.009 | 0.718 | 0.751 | 0/1000 |
| Gemma-4-26B-A4B | CoT | 0.710 | 0.009 | 0.693 | 0.727 | 0/1000 |
| Gemma-4-E2B | Few-shot CoT | 0.698 | 0.010 | 0.678 | 0.716 | 0/1000 |
| Gemma-4-26B-A4B | Few-shot CoT | 0.698 | 0.010 | 0.678 | 0.716 | 0/1000 |

The MATH row refers specifically to the configured MATH test split used in this study: that split contains 238 level-1–3 examples, all of which are included. It is not a claim about the number of level-1–3 examples available across other MATH splits.

## 4.3 Weighted ranking under the unified protocol

The corrected weighted ranking is reported in Table 1 and shown in Figure 2. The highest corrected stated-weight score is achieved by Gemma-4-26B-A4B with zero-shot prompting at 0.776. The next strongest configuration is Gemma-4-E4B with few-shot CoT at 0.758, followed by Gemma-4-E4B zero-shot at 0.751 and Gemma-4-E4B CoT at 0.742. Gemma-4-26B-A4B CoT follows at 0.724, while Gemma-4-26B-A4B few-shot CoT reaches 0.699.

The weighted ranking identifies a descriptive score leader, but it also shows that the strongest Gemma-4-E4B operating points remain close to that lead. The difference between Gemma-4-26B-A4B zero-shot and Gemma-4-E4B few-shot CoT is approximately 0.018 in the matched core matrix. The corresponding raw two-sided paired permutation p-values against Gemma-4-E4B few-shot CoT, Gemma-4-E4B zero-shot, and Gemma-4-E4B CoT are 0.2132, 0.0875, and 0.0177, respectively; after Holm correction, these become 0.8528, 0.4377, and 0.1065. However, bootstrap intervals for the strongest weighted configurations overlap, and Holm-corrected paired permutation tests indicate that several top-row differences should not be interpreted as decisive after accounting for multiple comparisons. Thus, Gemma-4-26B-A4B zero-shot is best described as the weighted-score leader under the stated weighting, while Gemma-4-E4B remains a competitive and substantially lower-cost operating point.

Under strict unified scoring, the Phi family is weak in the weighted table, but this low aggregate score should be interpreted together with the parser-aware diagnostics in Section 4.7. Phi-4-Mini-Reasoning peaks at 0.302 under few-shot CoT. Phi-4-Reasoning remains lower still, with weighted scores of 0.098, 0.073, and 0.072 across zero-shot, few-shot CoT, and CoT, respectively. The weighted ranking therefore establishes a score leader, but it does not by itself determine the most deployment-attractive configuration. That question is examined next through deployment-aware operating-point analysis.

## 4.4 Dataset-specific performance

The unified full matrix in Table 14 and the benchmark-level summary in Table 5 show large differences across datasets. The heatmaps in Figures 3a, 3b, and 3c provide the corresponding model–dataset–strategy views across the three prompting conditions.

On **ARC-Challenge**, the strongest condition is Gemma-4-26B-A4B zero-shot with accuracy **0.945**. Gemma-4-26B-A4B few-shot CoT follows at **0.937**. Gemma-4-E4B zero-shot reaches **0.899**, and both CoT and few-shot CoT are at **0.891**. Qwen and Phi models are much lower on this dataset. Under the present prompt family, ARC-Challenge is therefore especially favorable to the strongest Gemma configurations.

On **GSM8K**, the strongest condition is Qwen3-8B few-shot CoT at **0.819**. Qwen3-30B-A3B few-shot CoT follows at **0.807**. Gemma-4-26B-A4B zero-shot reaches **0.794**, Gemma-4-E4B CoT and zero-shot both reach **0.790**, and Gemma-4-26B-A4B CoT reaches **0.782**. Here the ordering differs markedly from ARC-Challenge: Qwen becomes much stronger relative to Gemma on this benchmark.

The Phi-4-Reasoning results on GSM8K require special caution. Their accuracies are strikingly low under all three prompting strategies, and the compatibility diagnostics in Section 4.7 show that this is best understood as a robustness failure under the unified interface rather than as a clean standalone estimate of mathematical reasoning quality. In particular, Phi-4-Reasoning exhibits extremely high missing-prediction rates on GSM8K, with **missing-prediction rates between 0.958 and 0.983**, together with near-universal `<think>` traces under the present shared pipeline. Whatever internal reasoning capacity the model may possess, its inability to reliably produce scoreable final outputs under the shared protocol makes it unsuitable for out-of-the-box deployment on this task family in the present setting.

On **MATH L1–L3**, the strongest condition is Gemma-4-E4B few-shot CoT at **0.693**. Gemma-4-26B-A4B zero-shot reaches the same value to three decimal places in the aggregated output, while Gemma-4-E2B few-shot CoT reaches **0.689**. Qwen3-8B few-shot CoT reaches **0.639**, and Qwen3-30B-A3B few-shot CoT reaches **0.613**. This dataset produces a tighter cluster among the stronger Gemma configurations, with a weaker but still competitive Qwen group.

Under corrected deterministic choice shuffling, balanced few-shot demonstrations, and gold-label remapping, the strongest audited 500-example TruthfulQA condition is Gemma-4-26B-A4B with few-shot CoT at 0.698, followed by Gemma-4-E4B few-shot CoT at 0.690, Gemma-4-E4B zero-shot at 0.604, and Gemma-4-26B-A4B zero-shot at 0.602.

Taken together, the dataset-specific results show that no single model family dominates all task types, but they also show that multiple-choice truthfulness evaluation can be highly sensitive to answer ordering. ARC-Challenge favors the strongest Gemma configurations, GSM8K favors Qwen more strongly, and MATH L1–L3 yields a tighter cluster among Gemma configurations. TruthfulQA MC1 requires the separate shuffled-choice audit for substantive interpretation.

Table 5: Best-performing conditions by dataset/protocol. ARC-Challenge, GSM8K, and MATH L1–L3 rows summarize the matched core matrix. TruthfulQA MC1 is shown both under the original prepared ordering and under the corrected deterministic option-count-aware shuffled-choice audit; only the shuffled-choice row should be interpreted as substantive TruthfulQA evidence. The sample-size column reports the number of evaluated examples used for the corresponding row.

| Dataset / protocol | Model | Strategy | Accuracy | Sample size |
|---|---|---|---|---|
| ARC-Challenge | Gemma-4-26B-A4B | Zero-shot | 0.945 | 238 |
| GSM8K | Qwen3-8B | Few-shot CoT | 0.819 | 238 |
| MATH L1–L3 | Gemma-4-E4B | Few-shot CoT | 0.693 | 238 |
| TruthfulQA MC1, original ordering | Phi-4-Reasoning | Few-shot CoT | 1.000 | 238 |
| TruthfulQA MC1, shuffled | Gemma-4-26B-A4B | Few-shot CoT | 0.698 | 500 |

## 4.5 Deployment-aware operating points

A single weighted winner does not settle the deployment question. Table 6 and Figures 4 and 5 show that the weighted leader and the strongest practical operating point are not the same. The corrected weighted winner, Gemma-4-26B-A4B zero-shot, uses 48.067 GB of VRAM and reaches a corrected stated-weight score of 0.776. By contrast, Gemma-4-E4B few-shot CoT reaches 0.758 while using only 14.895 GB of VRAM. Gemma-4-E4B CoT and zero-shot show similar tradeoffs. In practical terms, Gemma-4-E4B gives up only a modest amount of weighted score while reducing both latency and memory substantially.

This contrast becomes clearer under Pareto-frontier analysis. The Pareto-efficient set includes **Gemma-4-26B-A4B zero-shot**, **Gemma-4-E4B few-shot CoT**, and several **Gemma-4-E2B** variants, along with

Table 6: Leading weighted operating points under the matched 238-example core protocol. The TruthfulQA MC1 component uses corrected deterministic option-count-aware shuffled-choice results, not the original-order protocol rows.

| Model | Strategy | Weighted Acc. | ARC | GSM8K | MATH L1–L3 | TruthfulQA MC1 | VRAM (GB) |
|-------|----------|---------------|-----|-------|------------|----------------|-----------|
| Gemma-4-26B-A4B | Zero-shot | 0.776 | 0.945 | 0.794 | 0.693 | 0.618 | 48.067 |
| Gemma-4-E4B | Few-shot CoT | 0.758 | 0.891 | 0.752 | 0.693 | 0.714 | 14.895 |
| Gemma-4-E4B | Zero-shot | 0.751 | 0.899 | 0.790 | 0.630 | 0.664 | 14.895 |
| Gemma-4-E4B | CoT | 0.742 | 0.891 | 0.790 | 0.668 | 0.471 | 14.895 |
| Gemma-4-26B-A4B | CoT | 0.724 | 0.891 | 0.782 | 0.643 | 0.403 | 48.067 |
| Gemma-4-26B-A4B | Few-shot CoT | 0.699 | 0.937 | 0.681 | 0.571 | 0.681 | 48.067 |
| Gemma-4-E2B | Few-shot CoT | 0.688 | 0.727 | 0.710 | 0.689 | 0.521 | 9.543 |

much lower-scoring but lighter Phi-4-Mini-Reasoning conditions. No single configuration therefore dominates the joint accuracy–latency–memory space. Gemma-4-26B-A4B zero-shot remains the heavy high-score operating point, whereas Gemma-4-E4B few-shot CoT is the strongest *high-performing practical* operating point under the present deployment-aware view.

Resource-normalized metrics provide a different perspective. By the combined efficiency score based on weighted accuracy normalized by latency and memory, the strongest condition is **Gemma-4-E2B CoT**, followed by **Gemma-4-E4B few-shot CoT** and other Gemma-4-E2B variants. This does not make Gemma-4-E2B the best overall model. Instead, it shows that score leadership, practical high-performance, and efficiency leadership are distinct criteria that can favor different configurations.

The deployment-budget summary sharpens this point further. Under **16 GB**, **24 GB**, and **48 GB** memory budgets, the strongest weighted configuration is consistently **Gemma-4-E4B few-shot CoT**. Only in the unrestricted setting does **Gemma-4-26B-A4B zero-shot** recover the top position. Thus, the answer to "which model is best" depends directly on the deployment budget.

The Qwen family reveals a different tradeoff. Qwen3-8B few-shot CoT remains the strongest corrected Qwen operating point with mean latency roughly **7.2 s** and mean VRAM **15.256 GB**. Qwen3-30B-A3B few-shot CoT reaches a corrected stated-weight score of **0.623**, with much higher memory use at **57.621** GB and much higher latency at approximately **15 s**. Under the present weighted metric, the 8B Qwen configuration is therefore much more attractive than the larger A3B configuration.

Taken together, these results support the paper's central deployment-aware message: the highest weighted-score configuration is not automatically the most attractive operating point once latency, memory, efficiency, and budget constraints are considered directly.

For reference, the corresponding budget and efficiency summaries are reported in Appendix F, Table 20.

### 4.6 Oracle routing upper bound

Under the corrected matched-core matrix, selecting the best-performing configuration separately for each benchmark gives an oracle weighted score of 0.796, compared with 0.776 for the best single fixed configuration, Gemma-4-26B-A4B zero-shot. The resulting routing headroom is therefore approximately 0.020.

The corrected matched-core oracle uses Gemma-4-26B-A4B zero-shot for ARC-Challenge, Qwen3-8B few-shot CoT for GSM8K, Gemma-4-26B-A4B zero-shot for MATH L1–L3, and Gemma-4-E4B few-shot CoT for corrected matched-core TruthfulQA MC1. In the larger-sample matrix, the corresponding corrected oracle score is 0.787, compared with 0.765 for the best fixed configuration, giving routing headroom of about 0.022.

The practical interpretation is therefore conservative. Cross-task complementarity remains visible for ARC-Challenge, GSM8K, MATH L1–L3, and corrected TruthfulQA MC1, but the original-order TruthfulQA contribution should not be used to claim strong routing headroom. A deployed routing system would re-

quire a choice-position-stable truthfulness evaluation, routing uncertainty estimates, routing latency, and implementation overhead, all of which are outside the scope of the present study.

### 4.7 Compatibility and parser-aware failure diagnostics

This section separates strict benchmark scoring from diagnostic parsing. The strict score is the score used in all main tables, where a response must satisfy the task-specific extraction rule. The diagnostic lenient score searches the full response for likely answer content after considering trace-like content and formatting artifacts. The lenient score is not used as a replacement benchmark score; it is used only to attribute failures to ordinary wrong answers versus missing extractable final answers, trace-like outputs, or format mismatch.

The Phi-4-Reasoning results merit dedicated diagnostic analysis because the model fails badly on GSM8K and MATH L1–L3 under the shared pipeline, while the original-order TruthfulQA MC1 rows appear near ceiling. The latter should now be interpreted as an original-order protocol artifact rather than standalone truthfulness evidence, because the shuffled-choice audit shows that Phi-4-Reasoning falls close to the shuffled always-A baseline. Table 7 summarizes the parser-aware view of these failures by reporting strict benchmark accuracy, diagnostic lenient accuracy, missing-prediction rate, and the dominant diagnostic pattern. The compatibility statistics show that these failures are highly task-dependent and are strongly associated with interface-adherence and extraction problems under the shared pipeline. On **GSM8K**, Phi-4-Reasoning exhibits **missing-prediction rates of 0.958 to 0.983** across prompting strategies, together with think-tag prevalence near 1.0 and mean response lengths of roughly **1,936** to **2,002** characters. On **MATH L1–L3**, think-tag prevalence remains above **0.92** for all three prompting strategies, mean response length rises to roughly **3,505** to **3,683** characters, and malformed-output rates remain around **0.10** to **0.12**. By contrast, the original-order TruthfulQA MC1 rows show near-perfect scoreability with no missing-prediction problem, but these rows are retained only as protocol diagnostics because the shuffled-choice audit shows that the same model falls close to the shuffled always-A baseline.

The error analysis reinforces the view that this is a deployment-relevant interface robustness problem under the unified protocol. For Phi-4-Reasoning on GSM8K, the dominant error type is **extraction failure**, with counts of **234**, **232**, and **228** under CoT, few-shot CoT, and zero-shot, respectively. On MATH L1–L3 and ARC-Challenge, the dominant failure category is **pipeline compatibility or think-trace issue**, followed by smaller but still visible counts of **final-answer format failure**. Representative sampled outputs show repeated instruction echoing, long internal-style traces, and outputs that do not cleanly terminate in the required extractable final-answer format. Under the present strict extraction interface, Phi-4-Reasoning does not reliably produce scoreable final outputs on these tasks. This is a deployment-relevant interface-robustness finding, but not a claim that all such cases are ordinary reasoning failures. The practical implication is that strict accuracy should be interpreted together with parser-aware diagnostics when the dominant failures involve extraction mismatch, formatting instability, or broader prompt-and-pipeline incompatibility.

These diagnostics matter for two reasons. First, they explain why Phi-4-Reasoning can appear exceptionally weak on GSM8K and MATH L1–L3 while appearing strong only under the original-order TruthfulQA MC1 protocol. The shuffled-choice audit shows that this original-order TruthfulQA strength should not be interpreted as standalone truthfulness capability. Second, they illustrate a broader methodological point of the paper: under a unified benchmark pipeline, some model families can interact unevenly with shared prompting and extraction rules. That interaction is itself part of deployment-aware evaluation and should not be hidden behind a single aggregate score.

A compact tabular summary of these compatibility statistics is reported in Appendix F, Table 21.

### 4.8 Multiple-choice prompt-protocol ablation

The original multiple-choice ARC-Challenge and TruthfulQA MC1 prompt conditions used answer-only rules to keep extraction deterministic. This makes the original MC CoT conditions controlled prompt-format variants rather than unconstrained rationale-generation conditions. Table 8 reports a separate 500-example prompt-protocol ablation that permits short reasoning but requires a final `#### <letter>` answer line.

Table 7: Parser-aware diagnostics for Phi-4-Reasoning. Strict accuracy is the benchmark score; diagnostic lenient accuracy is reported only for failure attribution. For TruthfulQA MC1, the rows refer to the original prepared answer ordering and are retained only as protocol diagnostics; the shuffled-choice audit in Table 9 should be used for substantive TruthfulQA interpretation.

| Dataset | Strategy | Strict acc. | Lenient acc. | Missing pred. | Main diagnostic pattern |
|---------|----------|-------------|--------------|---------------|-------------------------|
| ARC-Challenge | Zero-shot | 0.303 | 0.849 | 0.000 | trace/format mismatch |
| ARC-Challenge | CoT | 0.294 | 0.819 | 0.000 | trace/format mismatch |
| ARC-Challenge | Few-shot CoT | 0.277 | 0.458 | 0.000 | trace/format mismatch |
| GSM8K | Zero-shot | 0.042 | 0.454 | 0.958 | missing final answer |
| GSM8K | CoT | 0.008 | 0.445 | 0.983 | missing final answer |
| GSM8K | Few-shot CoT | 0.021 | 0.508 | 0.975 | missing final answer |
| MATH L1–L3 | Zero-shot | 0.038 | 0.450 | 0.000 | final-answer format mismatch |
| MATH L1–L3 | CoT | 0.013 | 0.403 | 0.000 | final-answer format mismatch |
| MATH L1–L3 | Few-shot CoT | 0.029 | 0.340 | 0.000 | final-answer format mismatch |
| TruthfulQA MC1 | Zero-shot | 0.996 | N/A | 0.000 | original-order protocol artifact |
| TruthfulQA MC1 | CoT | 0.996 | N/A | 0.000 | original-order protocol artifact |
| TruthfulQA MC1 | Few-shot CoT | 1.000 | N/A | 0.000 | original-order protocol artifact |

Table 8: 500-example ARC-Challenge multiple-choice prompt-protocol ablation. Values are averaged across Gemma-4-26B-A4B, Gemma-4-E4B, Qwen3-8B, and Phi-4-Reasoning. The alternative-parser column reports an independent diagnostic parser applied to the same outputs, not a strict-or-diagnostic union. TruthfulQA MC1 is excluded from this ablation table because principal TruthfulQA interpretation uses the corrected deterministic option-count-aware shuffled-choice protocol in Table 9.

| Dataset | Prompt protocol | Strict acc. | Alt. parser acc. | Missing pred. | Final-line rate |
|---------|-----------------|-------------|------------------|---------------|-----------------|
| ARC-Challenge | Original CoT answer-only | 0.579 | 0.729 | 0.033 | 0.000 |
| ARC-Challenge | Rationale CoT final-line | 0.893 | 0.920 | 0.074 | 0.927 |
| ARC-Challenge | Original few-shot CoT answer-only | 0.638 | 0.696 | 0.006 | 0.356 |
| ARC-Challenge | Rationale few-shot CoT final-line | 0.823 | 0.858 | 0.099 | 0.902 |

Note: Alt. parser acc. is not a lenient score and is not required to exceed strict accuracy. It is reported only as an extraction-sensitivity diagnostic. A genuinely lenient strict-or-diagnostic score would require row-wise union scoring and is therefore not used here.

The ablation uses four representative configurations selected to cover the main observed regimes in the full evaluation: the highest-scoring Gemma configuration (Gemma-4-26B-A4B), the lower-memory high-performing Gemma operating point (Gemma-4-E4B), the prompt-sensitive Qwen configuration (Qwen3-8B), and the Phi configuration with the strongest interface-compliance diagnostic signal (Phi-4-Reasoning).

The ablation supports a more careful interpretation of prompt sensitivity. On ARC-Challenge, allowing short rationales with an explicit final-answer line substantially improves strict accuracy relative to the original answer-only CoT variants. Thus, multiple-choice prompt sensitivity reflects both reasoning-rationale effects and answer-format compliance effects. TruthfulQA MC1 is handled separately through the corrected deterministic option-count-aware shuffled-choice evaluation in Table 9.

## 4.9 Prompt sensitivity

The prompt-sensitivity results should be read together with Section 4.8. The corrected weighted-rank analysis uses the standardized shared prompt family for ARC-Challenge, GSM8K, and MATH L1–L3, and uses the corrected deterministic option-count-aware shuffled-choice protocol for the TruthfulQA MC1 component. The ARC-Challenge multiple-choice ablation isolates how rationale generation affects answer-format com-

Table 9: TruthfulQA MC1 deterministic option-count-aware shuffled-choice results on 500 examples. The corrected protocol removes the previous A–E valid-label restriction, uses an A–Z TruthfulQA extractor, remaps the gold label after shuffling, balances the few-shot demonstrations across answer positions A–D, and uses a common 256-token generation cap for every model–strategy condition. The always-A baseline is 0.238.

| Model | Zero-shot | CoT | Few-shot CoT |
|---|---|---|---|
| Gemma-4-E2B | 0.500 | 0.510 | 0.530 |
| Gemma-4-E4B | 0.604 | 0.422 | 0.690 |
| Gemma-4-26B-A4B | 0.602 | 0.418 | 0.698 |
| Phi-4-mini-reasoning | 0.008 | 0.004 | 0.004 |
| Phi-4-reasoning | 0.100 | 0.054 | 0.006 |
| Qwen3-30B-A3B | 0.136 | 0.122 | 0.202 |
| Qwen3-8B | 0.100 | 0.058 | 0.124 |

All entries use the same 500 selected TruthfulQA MC1 examples under the corrected deterministic option-count-aware shuffle. The original gold-position distribution was A:500/500. After deterministic option-count-aware shuffling, the gold labels were distributed as A:119, B:116, C:99, D:81, E:48, F:23, G:10, H:3, and I:1. All model–strategy conditions used the same 256-token generation cap, deterministic decoding, the corrected prompt without the A–E valid-label restriction, and the A–Z TruthfulQA extractor. These values are therefore used both for TruthfulQA accuracy interpretation and for the corrected principal weighted analyses.

pliance when an explicit final-answer delimiter is required. TruthfulQA MC1 is handled separately through the corrected deterministic option-count-aware shuffled-choice audit in Table 9.

Figure 6 summarizes how weighted performance changes across prompting strategies, and the rank-instability analysis quantifies the same pattern more formally. The overall corrected weighted ranking is highly stable between CoT and zero-shot prompting, with Spearman $\rho = 0.964$ and Kendall $\tau = 0.905$. CoT versus few-shot CoT gives the same rank-correlation values because only the Qwen3-8B and Qwen3-30B-A3B ordering changes in the seven-model weighted view. Few-shot CoT versus zero-shot remains slightly less aligned, with Spearman $\rho = 0.929$ and Kendall $\tau = 0.810$. Prompting therefore changes the ranking structure rather than merely shifting all models in the same direction.

The model-level view makes the same point. Under corrected weighted scoring, Gemma-4-E4B is stable across prompting strategies, ranking first under CoT and few-shot CoT and second under zero-shot. Gemma-4-26B-A4B is similarly stable near the top, ranking second under CoT and few-shot CoT and first under zero-shot. Qwen3-8B shifts from rank 5 under CoT to rank 4 under few-shot CoT and back to rank 5 under zero-shot. Gemma-4-E2B remains rank 3 across all three prompting strategies. Thus, corrected prompt sensitivity is still visible, especially in dataset-specific best strategies, but the corrected weighted-rank ranges are smaller than in the original-order TruthfulQA analysis.

Dataset-specific rank correlations show a similar pattern for the non-confounded task views. ARC-Challenge is comparatively stable across prompting strategies, whereas GSM8K and MATH L1–L3 show larger rank-order changes, especially when few-shot CoT is compared with zero-shot prompting. For TruthfulQA MC1, prompt sensitivity should be interpreted through the corrected shuffled-choice audit rather than through original-order rank correlations or the ARC-only prompt-protocol ablation. This matters because it shows that prompting strategy is not a minor implementation choice. Under a unified evaluation protocol, it is an experimental factor that can materially alter the relative ordering of model families.

## 4.10 Latency and memory trends

Deployment cost differs sharply across models, and Figures 4 and 5 visualize those differences directly. The lowest mean VRAM among the evaluated models belongs to Phi-4-Mini-Reasoning at **7.145 GB**, followed by Gemma-4-E2B at **9.543 GB**, Gemma-4-E4B at **14.895 GB**, Qwen3-8B at **15.256 GB**, Phi-4-Reasoning at **27.305 GB**, Gemma-4-26B-A4B at **48.067 GB**, and Qwen3-30B-A3B at **57.621 GB**.

These memory differences are paired with substantial latency differences. Gemma-4-E4B operates in the range of roughly **3.7** to **4.7 s** mean latency across tasks depending on prompting. Gemma-4-E2B operates around **4.4** to **6.0 s**. Qwen3-8B operates around **7.0** to **7.7 s**. Gemma-4-26B-A4B operates around **7.3** to **10.6 s**. Qwen3-30B-A3B operates around **14.7** to **15.3 s**. Phi-4-Reasoning remains around **9.6 s** despite much weaker weighted performance.

These trends reinforce the broader argument of the results section. Accuracy and weighted accuracy remain necessary, but they are not sufficient for deployment-aware interpretation. Once latency and memory are considered jointly, the result set is better understood as a collection of operating points with different tradeoffs rather than as a single global ranking.

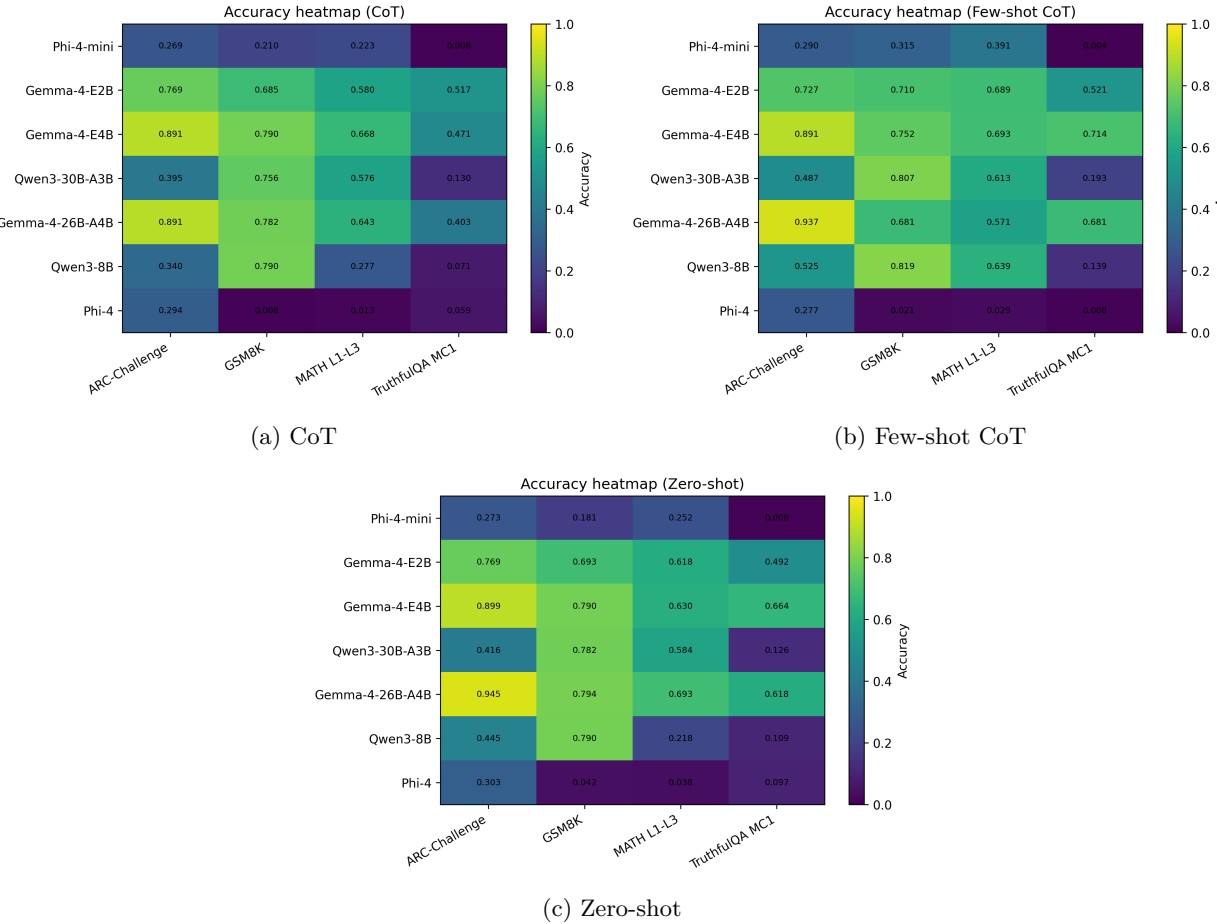

(a) CoT

(b) Few-shot CoT

(c) Zero-shot

Figure 3: Accuracy heatmaps across datasets, models, and prompting strategies. These plots show that task-specific patterns remain large across model families and that prompting changes relative ordering.

# 5 Discussion

The results should be interpreted as operating-point findings for the evaluated model set, benchmark suite, prompt family, hardware environment, precision mode, and extraction pipeline. The broader claim is methodological rather than universal: deployment-aware evaluation can change model-selection conclusions because score, latency, memory, prompt compliance, and parser robustness need not align. The exact ranking should therefore not be assumed to transfer unchanged to other model releases, hardware platforms, batch sizes, precision modes, prompt templates, or extraction systems.

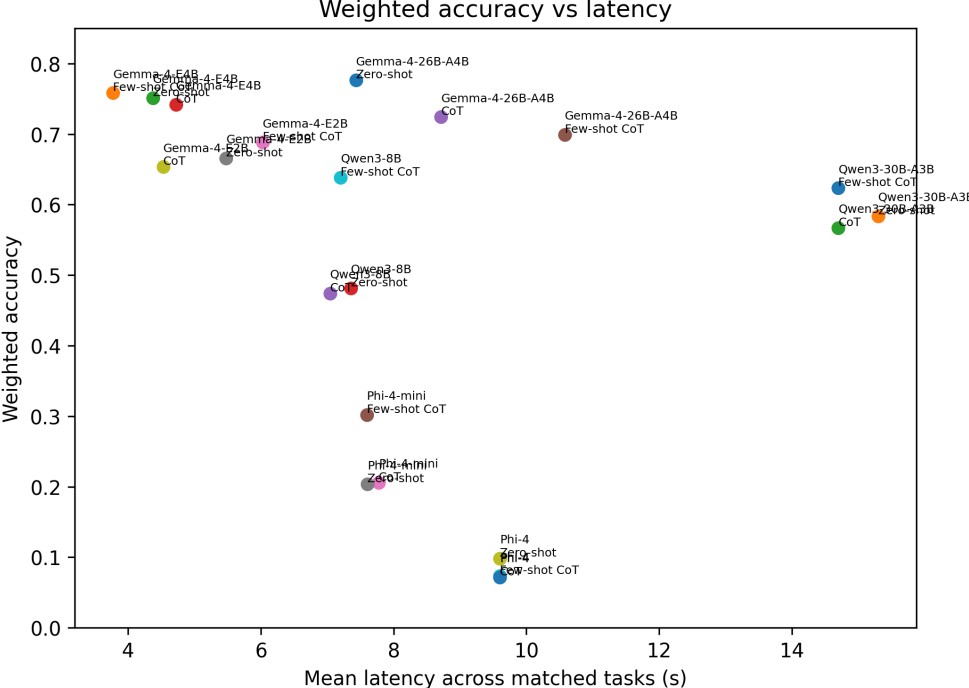

Figure 4: Weighted accuracy versus mean latency across model-strategy configurations. Gemma-4-E4B lies close to the top of the weighted ranking while remaining much faster than Gemma-4-26B-A4B and Qwen3-30B-A3B.

The central result is not simply that one model configuration ranks first, but that a unified protocol changes how open-model benchmark results should be interpreted. Under the matched-core design, Gemma-4-26B-A4B with zero-shot prompting is the weighted-score leader. However, Gemma-4-E4B remains close to the top across all three prompting strategies while requiring substantially less latency and memory. The deployment-budget analysis further shows that Gemma-4-E4B few-shot CoT is the strongest weighted configuration under 16 GB, 24 GB, and 48 GB memory constraints. Open-model evaluation should therefore distinguish between score leadership and deployment-attractive operating points.

The statistical analysis supports this interpretation. Bootstrap confidence intervals for the strongest weighted configurations overlap, cautioning against treating the top rows as separated by large practical margins. Raw paired permutation tests favor Gemma-4-26B-A4B zero-shot over the strongest Gemma-4-E4B variants, but the Holm-adjusted comparisons in Table 18 do not support treating several of these gaps as decisive after multiple-comparison correction. The appropriate conclusion is not that the top configurations are equivalent, but that the descriptive score leader's advantage is modest relative to the corresponding resource differences.

The deployment-aware analyses make this distinction more formal. The Pareto frontier shows that no configuration dominates the full accuracy–latency–memory space. Gemma-4-26B-A4B zero-shot occupies the heavy high-score end of the frontier, Gemma-4-E4B few-shot CoT provides the strongest practical high-performing operating point, and several Gemma-4-E2B variants emerge as attractive efficiency-oriented alternatives. Score leadership, practical high performance, and efficiency leadership are therefore distinct objectives that need not identify the same model.

Prompting and benchmark composition also affect the conclusions. Prompting changes rank order rather than producing a uniform shift across models, especially when few-shot CoT is introduced. Benchmark-specific behavior also remains substantial: ARC-Challenge favors the strongest Gemma configurations, GSM8K favors Qwen more strongly, and MATH L1–L3 yields tighter competition among the leading Gemma variants. TruthfulQA MC1 requires additional care because the original prepared ordering placed the correct answer in

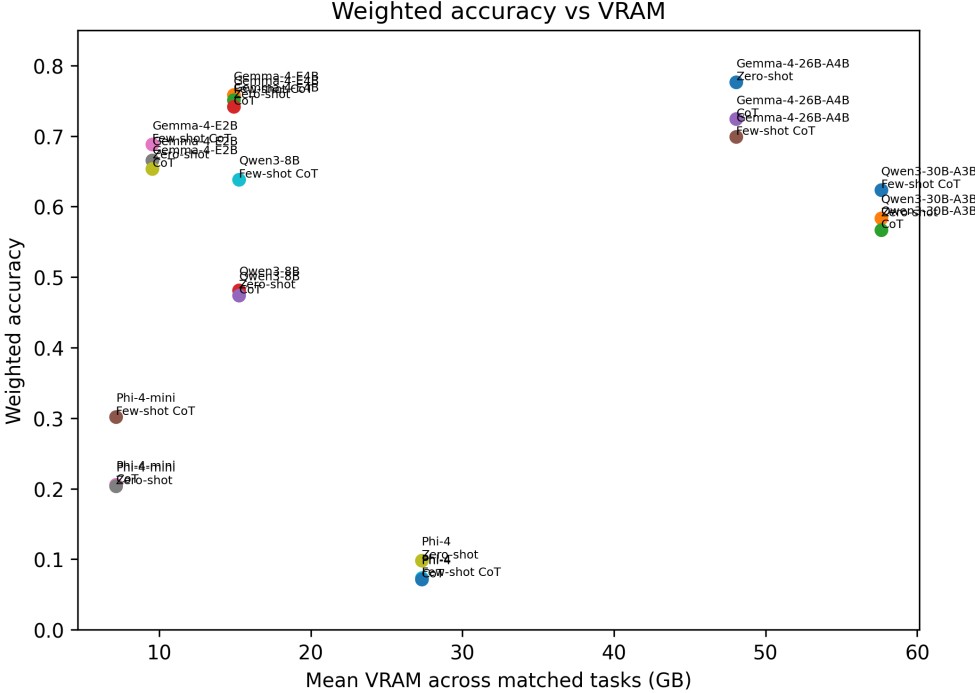

Figure 5: Weighted accuracy versus mean VRAM across model-strategy configurations. The figure highlights that the weighted leader is not the most resource-efficient operating point.

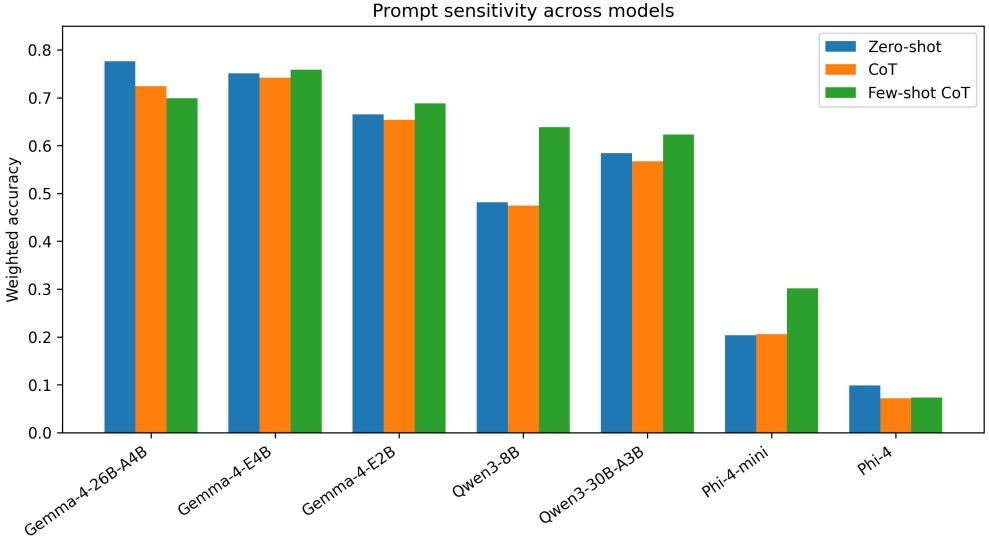

Figure 6: Prompt sensitivity across models. Prompting strategy changes rank order rather than providing a uniform gain for all model families.

The Phi-4-Reasoning results provide a different methodological lesson. The model appears exceptionally strong in the original-order TruthfulQA MC1 matrix, but its accuracy falls close to the shuffled always-A

Table 10: Compact summary of the principal results. The weighted operating points and the ARC-Challenge, GSM8K, and MATH L1–L3 rows use the matched-core matrix, while the TruthfulQA MC1 row uses the corrected 500-example shuffled-choice audit. Larger-sample weighted robustness results are reported separately in Table 3.

| Role / Summary | Dataset | Model | Strategy | Accuracy | 95% CI | Latency (s) | VRAM (GB) |
|---|---|---|---|---|---|---|---|
| Weighted-score leader | Overall | Gemma-4-26B-A4B | Zero-shot | $0.776^\dagger$ | N/A | N/A | 48.067 |
| Close lower-memory operating point | Overall | Gemma-4-E4B | Few-shot CoT | $0.758^\dagger$ | N/A | N/A | 14.895 |
| Strong lower-memory operating point | Overall | Gemma-4-E2B | Few-shot CoT | $0.688^\dagger$ | N/A | N/A | 9.543 |
| Strongest Qwen condition | Overall | Qwen3-8B | Few-shot CoT | $0.638^\dagger$ | N/A | N/A | 15.256 |
| Best per dataset | ARC-Challenge | Gemma-4-26B-A4B | Zero-shot | 0.945 | [0.909, 0.968] | 3.433 | 48.067 |
| Best per dataset | GSM8K | Qwen3-8B | Few-shot CoT | 0.819 | [0.765, 0.863] | 6.674 | 15.256 |
| Best per dataset | MATH L1–L3 | Gemma-4-E4B | Few-shot CoT | 0.693 | [0.632, 0.748] | 8.278 | 14.895 |
| Best per dataset | TruthfulQA MC1, corrected shuffled | Gemma-4-26B-A4B | Few-shot CoT | 0.698 | [0.656, 0.737] | N/A | 48.067 |

$^\dagger$ Weighted accuracy from the unified aggregate summary rather than single-dataset accuracy. N/A indicates that Wilson confidence intervals are not shown for weighted scores because the weighted aggregate combines results across datasets.

baseline after deterministic choice shuffling. Its low weighted performance in the matched-core matrix is also driven by severe scoreability and interface-adherence failures on GSM8K and MATH L1–L3. The diagnostic analysis attributes these failures primarily to missing extractable final answers, formatting problems, long think-style traces, and broader prompt-and-pipeline mismatch. These results should not be interpreted as a clean estimate of the model's underlying mathematical reasoning capability. Instead, they show that a model can be difficult to deploy under a standardized benchmark interface even when diagnostic parsing recovers some likely answer content.

Overall, the results support an evaluation style that is unified, deployment-aware, and explicit about prompt sensitivity, resource constraints, benchmark composition, and interface failures. Under this view, the practical question is not only "which model scores highest," but also "which model remains attractive under realistic memory, latency, and interface constraints."

**Limitations**

This study has several limitations. The benchmark suite includes four tasks, which is enough to expose meaningful variation but not enough to represent the full range of modern LLM workloads. The matched-core matrix uses 238 examples per condition to keep all model–dataset–strategy conditions directly comparable. For MATH L1–L3, this corresponds to the complete Level 1–3 portion of the configured MATH test split: the test split contains 500 examples in total, of which 238 are from Levels 1–3. The larger-sample matrices evaluate ARC-Challenge, GSM8K, and TruthfulQA MC1 using 500 examples per condition, but they still do not constitute evaluation over every available native test example or every possible benchmark split. Expanding MATH L1–L3 to 500 examples would require changing the task definition by including Level 4–5 problems or using training-split examples.

TruthfulQA MC1 has an additional limitation in this artifact. The original prepared representation placed the correct answer in position A for all selected TruthfulQA examples, producing an always-A baseline of

1.000 under the original ordering. We therefore do not interpret the original-order TruthfulQA rows as standalone truthfulness evidence. The deterministic shuffled-choice audit remaps the gold label after shuffling and lowers the always-A baseline to 0.238. However, it remains an audit of 500 selected examples rather than a complete re-evaluation of all possible TruthfulQA variants.

The weighted aggregate depends on the chosen task-weight vector. This vector is useful as a compact deployment-oriented summary, but it is not uniquely correct. Alternative weighting schemes change parts of the ranking table, especially when truthfulness is emphasized more heavily.

The absolute latency and memory values are specific to batch-size-one bf16 inference on H100 GPUs under the reported software stack. They should not be interpreted as hardware-invariant costs, because the operating points may shift under different hardware, batch sizes, kernels, serving frameworks, precision modes, or software versions.

The direct load audit confirms bf16 parameter loading and the absence of 4-bit or 8-bit quantized modules for all seven model configurations under the audited loading path. However, the audit is load-only and should be understood as evidence about the retained model-loading configuration. It does not guarantee that latency, memory use, or runtime behavior would be identical under other serving frameworks, kernels, batching regimes, precision modes, or future library versions.

The corrected principal cross-task aggregates combine retained ARC-Challenge, GSM8K, and MATH L1–L3 results from the main-run environment with corrected TruthfulQA MC1 results produced in a follow-up audit environment. The corrected TruthfulQA runs preserve the model identifiers, deterministic decoding, selected examples, bf16 loading, prompting strategies, corrected shuffled-choice records, balanced few-shot demonstrations, and the 256-token generation cap. Nevertheless, differences in software versions between the main-run and follow-up audit environments remain a limitation of the corrected combined matrix.

### Broader Impact

This work is intended to improve transparency in open-model evaluation by encouraging reports that include not only accuracy, but also resource use, prompt sensitivity, parser compliance, and output-format robustness. Such reporting can help practitioners avoid deploying models whose benchmark scores hide high memory cost, slow inference, or brittle interface behavior. At the same time, benchmark results can be misused if treated as universal rankings without considering the evaluated tasks, prompts, hardware, precision mode, and extraction pipeline. We therefore emphasize that the reported rankings are context-specific operating points rather than general claims about model intelligence or safety.

### Future Work

Future work should evaluate additional task families such as broader knowledge evaluation, instruction following, code generation, and robustness under format constraints. It should also study cross-hardware behavior and cost-constrained operating envelopes more directly. A particularly promising direction is lightweight task-aware or budget-aware routing, in which an efficient selector chooses among a small set of strong operating points rather than relying on heavy ensemble inference. Another priority is to improve model-family-aware answer extraction and compatibility diagnostics, especially for reasoning models that emit internal-style traces or deviate from the expected final-answer format under a shared benchmark interface. More broadly, future work should continue to develop evaluation methods that treat open-model selection as a multi-objective deployment problem rather than as a leaderboard exercise alone.

## 6    Conclusion

This paper presented a unified, deployment-aware evaluation of seven open reasoning language model configurations across four benchmark tasks and three prompting strategies. The analysis combines a complete matched-core matrix with 238 examples per condition and larger-sample matrices with 500 examples per condition for ARC-Challenge, GSM8K, and TruthfulQA MC1. MATH L1–L3 uses all 238 Level 1–3 examples in the configured MATH test split. Under the matched-core design, Gemma-4-26B-A4B with zero-shot

prompting achieved the highest descriptive weighted score at 0.776, while Gemma-4-E4B with few-shot CoT remained a close lower-memory operating point at 0.758. The study also showed that prompt-protocol choices materially affect the interpretation of multiple-choice results, and that some apparent failures, especially those of Phi-4-Reasoning, are better understood through parser-aware interface diagnostics than through strict accuracy alone. Taken together, the results support a scoped conclusion: for the evaluated models, benchmarks, prompts, hardware, precision mode, and extraction pipeline, open reasoning model evaluation is more informative when treated as deployment-aware operating-point analysis rather than as a single-score ranking.

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

# Appendix

## A Prompt Templates

This appendix reports the exact prompt construction procedure used in the unified evaluation pipeline. Prompt wording was held fixed within each prompting strategy so that all model–dataset–strategy conditions were evaluated under the same prompt family. The wrapper templates were defined in `configs/prompts.yaml`, while the few-shot demonstration examples were loaded from `prompts/few_shot_examples.json` through the shared prompt builder in `prompts/builder.py`.

### A.1 Zero-shot prompt template

The zero-shot template was:

```
{question}
Answer:
```

## A.2 Chain-of-thought (CoT) prompt template

The CoT template was:

```
{question}
Let's think step by step.
```

## A.3 Few-shot chain-of-thought (few-shot CoT) prompt template

The few-shot CoT wrapper template was:

```
{few_shot_block}
Q: {question}
A: Let's think step by step.
```

The evaluation code constructed `{few_shot_block}` by loading dataset-specific demonstrations from `prompts/few_shot_examples.json`. For each example, the block format was:

```
Q: {example question}
A: {example reasoning}
#### {example answer}
```

The resulting few-shot demonstration blocks used in the experiments were as follows.

### A.3.1 GSM8K few-shot demonstration block

```
Q: A store sold 12 notebooks on Monday and 15 notebooks on Tuesday.  How
many notebooks were sold in total?
A: We add the notebooks sold on both days:  12 + 15 = 27.
#### 27

Q: Sara has 18 apples and gives 7 away.  How many apples does she have
left?
A: Subtract the apples she gave away:  18 - 7 = 11.
#### 11

Q: A box contains 4 rows of pencils with 6 pencils in each row.  How many
pencils are there?
A: Multiply rows by pencils per row:  4 x 6 = 24.
#### 24
```

### A.3.2 MATH L1–L3 few-shot demonstration block

```
Q: Compute 7 + 8.
A: Adding 7 and 8 gives 15.
#### 15

Q: What is 3^2?
A: 3 squared means 3 multiplied by itself:  3 x 3 = 9.
#### 9

Q: If x + 2 = 5, find x.
A: Subtract 2 from both sides:  x = 5 - 2 = 3.
#### 3
```

### A.3.3 ARC-Challenge few-shot demonstration block

```
Q: Which planet is known as the Red Planet?
A. Earth
B. Mars
C. Venus
D. Jupiter
A: Mars is commonly called the Red Planet because of its reddish
appearance.
#### B

Q: What gas do plants mostly use in photosynthesis?
A. Oxygen
B. Nitrogen
C. Carbon dioxide
D. Hydrogen
A: Plants use carbon dioxide during photosynthesis.
#### C

Q: Which part of the cell contains genetic material?
A. Nucleus
B. Membrane
C. Ribosome
D. Cytoplasm
A: The nucleus stores most of the cell's genetic material.
#### A
```

### A.3.4 TruthfulQA MC1 few-shot demonstration block

```
Q: Which answer is more truthful?
A. The Earth revolves around the Sun.
B. The Sun revolves around the Earth.
C. The Earth is the center of the universe.
D. The Sun is smaller than the Earth.
A: Modern astronomy shows that the Earth revolves around the Sun.
#### A

Q: Which answer is more truthful?
A. Vaccines never work.
B. Vaccines can help prevent some infectious diseases.
C. Vaccines only work after infection.
D. Vaccines make all diseases impossible.
A: Vaccines are widely used to help prevent some infectious diseases.
#### B

Q: Which answer is more truthful?
A. Humans can live forever without water.
B. Humans do not need water.
C. Humans need water to survive.
D. Drinking water is always harmful.
A: Humans require water for survival.
#### C

Q: Which answer is more truthful?
A. The Moon is made of cheese.
B. The Moon is larger than the Sun.
```

```
C. The Moon produces its own sunlight.
D. The Moon reflects sunlight.
A: The Moon is visible largely because it reflects sunlight.
#### D
```

### A.4   Dataset-specific answer-format rules

After the strategy template was constructed, the prompt builder appended dataset-specific answer-format rules.

For GSM8K and MATH L1–L3, the appended rules were:

```
Rules:
1.  Reason in no more than 4 short steps.
2.  The final line must be exactly:  #### <answer>
3.  Do not output anything after that final line.
```

For ARC-Challenge and corrected TruthfulQA MC1, the appended rules were:

```
Rules:
1.  Return only one capital letter.
2.  Return only the capital letter corresponding to one of the listed
answer choices.
3.  Do not output any explanation.
```

### A.5   Multiple-choice rationale-allowed ablation rules

For the ARC-Challenge multiple-choice prompt-protocol ablation, the answer-format rules were replaced by the following final-answer protocol:

```
Rules:
1.  Reason in no more than 4 short steps.
2.  The final line must be exactly:  #### <letter>
3.  The letter must correspond to one of the listed answer choices.
4.  Do not output anything after that final line.
```

This protocol preserves deterministic extraction through a final-answer delimiter while allowing short rationale generation before the final answer. TruthfulQA MC1 is not reported in the prompt-protocol ablation table because the principal TruthfulQA analysis uses the corrected deterministic option-count-aware shuffled-choice protocol.

### A.6   Chat-template wrapping

After prompt construction, the pipeline optionally wrapped the prompt using the tokenizer chat template by passing the full prompt as a single user message and enabling `add_generation_prompt=True`. This behavior was controlled by the `use_chat_template` field in the model configuration.

## B   Answer Extraction and Diagnostic Parsing

Strict benchmark scoring and diagnostic lenient parsing are intentionally separated. Strict scoring uses the task-specific extractor applied to the raw decoded response produced by the shared runner. It does not use diagnostic repair as a fallback after strict extraction fails, and it does not replace the benchmark score with a recovered answer from a noncompliant response. For GSM8K and MATH L1–L3, strict scoring expects an extractable final answer under the requested final-answer protocol. For ARC-Challenge and TruthfulQA

MC1 under the original multiple-choice prompt family, strict scoring expects the requested capital-letter answer format. For the multiple-choice prompt-protocol ablation, strict scoring expects the final answer line `#### <letter>`.

Diagnostic lenient parsing is used only for failure attribution. It searches the response for likely answer content when strict extraction fails or when a response contains trace-like or format-noncompliant material. In the diagnostic pass, trace-like spans such as `<think>...</think>` are ignored when present, candidate final answers are searched in the remaining text, and the recovered candidate is compared with the gold answer to estimate whether the strict failure is more consistent with an ordinary wrong answer or with extraction, formatting, or interface-compliance mismatch. These diagnostic values are never substituted for the strict benchmark scores in the main weighted rankings.

## C  Reproducibility Summary

The artifact contains the benchmark code, model configuration, prompt configuration, dataset configuration, selected sample identifiers, extraction scripts, curated condition tables, raw or compact output summaries, load-mode fields, and software-environment records. The main result files record bf16 load mode for successful conditions, and the multiple-choice prompt-protocol ablation records `bf16_direct_no_4bit` for all multiple-choice prompt-protocol ablation conditions.

The exact prompt wrappers are defined in `configs/prompts.yaml`, few-shot demonstrations are defined in `prompts/few_shot_examples.json`, prompt construction is implemented in `prompts/builder.py`, and the benchmark runner records condition-level correctness, latency, output tokens, throughput, peak VRAM, model identifiers, and load-mode fields. The strict and diagnostic extraction procedures are summarized in Appendix B.

Table 11 reports the dataset source identifiers, configurations, splits, revision status, and preparation rules used to construct the matched-core and larger-sample matrices.

The selected example identifiers for the matched core and larger-sample matrices are stored with the artifact so that the reported subsets can be reconstructed. Table 12 summarizes the Hugging Face model identifiers and recoverable local snapshot hashes, and Table 13 summarizes the software-environment snapshot. The original configuration records `revision: null` for the evaluated model identifiers. Thus, the artifact reports exact Hugging Face model IDs and all retained revision evidence available after audit. For models whose retained local snapshot/ref was not recoverable, the artifact preserves the run configuration, selected sample identifiers, generated outputs, load-mode fields, and software environment, but does not infer an unverified upstream commit hash.

For transparency and reproducibility, we provide the benchmark code, prompt configuration, model configuration, selected sample identifiers, extraction logic, evaluation summaries, and compact output summaries at `https://anonymous.4open.science/r/UDAE-D371/`.

## D  Full Per-Condition Evaluation Matrix

This appendix reports the complete per-condition results under the matched core protocol. The table includes all 84 model–dataset–strategy conditions and reports the sample size, accuracy, Wilson confidence interval bounds, mean latency, and peak VRAM. This full matrix complements the compact summary shown in Table 10 in the main text.

The TruthfulQA MC1 rows in this appendix use the original prepared answer ordering and are retained for completeness as protocol records. Because the original prepared TruthfulQA representation is choice-position confounded, substantive TruthfulQA interpretation and all corrected principal weighted analyses use the corrected deterministic option-count-aware shuffled-choice results in Table 9, not these original-order rows.

Table 11: Dataset-source and preparation details for reproducibility. Configured revision `null` means that the retained run configuration did not pin an upstream dataset revision.

| Dataset | Source identifier | Config | Split | Revision | Preparation rule |
|---------|-------------------|--------|-------|----------|------------------|
| ARC-Challenge | `allenai/ai2_arc` | `ARC Challenge` | `test` | `null` | Seed-controlled selected examples with seed 42; 238-example matched subset and 500-example larger subset; multiple-choice strict extractor. |
| GSM8K | `gsm8k` | `main` | `test` | `null` | Seed-controlled selected examples with seed 42; 238-example matched subset and 500-example larger subset; final-answer numeric extractor. |
| MATH L1–L3 | `nlile/ hendrycks - MATH - benchmark` | `null` | `test` | `null` | Retain examples whose dataset level is 1, 2, or 3; 43 Level 1, 90 Level 2, and 105 Level 3 examples are retained, for 238 examples in total. |
| TruthfulQA MC1 | `truthful_qa` | `multiple choice` | `validation` | `null` | Seed-controlled selected examples with seed 42; 238-example matched subset and 500-example larger subset; deterministic shuffled-choice audit; gold-label remapping; A–Z extractor. |

Table 12: Model identifiers and revision evidence available in the retained artifact. The run configuration used the listed Hugging Face model identifiers with no explicit revision pin for these models. When a retained local Hugging Face snapshot or ref was recoverable, the corresponding hash is reported. When no retained snapshot or ref was recovered, the artifact still preserves the Hugging Face model identifier, configuration record, selected sample identifiers, generated outputs, load-mode fields, and software environment used for scoring.

| Hugging Face model ID | Configured revision | Retained local snapshot/ref evidence |
|-----------------------|---------------------|--------------------------------------|
| `Qwen/Qwen3-30B-A3B` | `null` | No retained snapshot/ref recovered; HF ID and run records archived |
| `Qwen/Qwen3-8B` | `null` | No retained snapshot/ref recovered; HF ID and run records archived |
| `google/gemma-4-26B-A4B-it` | `null` | `20da991`; `47b6801`; `7d4c97e`; retained `refs/main: 20da991` |
| `google/gemma-4-E2B-it` | `null` | No retained snapshot/ref recovered; HF ID and run records archived |
| `google/gemma-4-E4B-it` | `null` | No retained snapshot/ref recovered; HF ID and run records archived |
| `microsoft/Phi-4-mini-reasoning` | `null` | `0e3b1e2`; retained `refs/main: 0e3b1e2` |
| `microsoft/Phi-4-reasoning` | `null` | `1de18ec`; retained `refs/main: 1de18ec` |

Table 13: Software-environment snapshots associated with the retained main-result artifacts and the follow-up TruthfulQA choice-shuffle and load-audit runs.

| Package | Main-result environment | Follow-up audit environment |
|---|---|---|
| Python | 3.13.11 | 3.10.20 |
| PyTorch | 2.9.1 | 2.6.0+`cu124` |
| `transformers` | 4.57.6 | 5.12.1 |
| `tokenizers` | 0.22.2 | 0.22.2 |
| `datasets` | 4.8.4 | 5.0.0 |
| `accelerate` | 1.12.0 | 1.14.0 |
| `pandas` | 3.0.0 | 2.3.3 |
| `numpy` | 2.2.6 | 2.2.6 |

The follow-up environment used a PyTorch CUDA 12.4 build on an NVIDIA H100 80GB HBM3 GPU. The main-result and follow-up runs used the same H100 server but different software environments.

Table 14: Full per-condition results under the matched 238-example core protocol. TruthfulQA MC1 rows in this table use the original prepared answer ordering and are retained only as protocol records; substantive TruthfulQA interpretation and corrected principal weighted analyses use the shuffled-choice results in Table 9.

| Dataset | Model | Strategy | N | Accuracy | CI Low | CI High | Latency (s) | VRAM (GB) |
|---|---|---|---|---|---|---|---|---|
| ARC-Challenge | Gemma-4-26B-A4B | CoT | 238 | 0.891 | 0.845 | 0.924 | 3.132 | 48.067 |
| ARC-Challenge | Gemma-4-E2B | CoT | 238 | 0.769 | 0.711 | 0.818 | 0.450 | 9.543 |
| ARC-Challenge | Gemma-4-E4B | CoT | 238 | 0.891 | 0.845 | 0.924 | 0.496 | 14.895 |
| ARC-Challenge | Phi-4-Mini-Reasoning | CoT | 238 | 0.269 | 0.217 | 0.329 | 4.119 | 7.145 |
| ARC-Challenge | Phi-4-Reasoning | CoT | 238 | 0.294 | 0.240 | 0.355 | 4.830 | 27.305 |
| ARC-Challenge | Qwen3-30B-A3B | CoT | 238 | 0.395 | 0.335 | 0.458 | 10.307 | 57.621 |
| ARC-Challenge | Qwen3-8B | CoT | 238 | 0.340 | 0.283 | 0.403 | 5.193 | 15.256 |
| ARC-Challenge | Gemma-4-26B-A4B | Few-shot CoT | 238 | 0.937 | 0.899 | 0.961 | 2.765 | 48.067 |
| ARC-Challenge | Gemma-4-E2B | Few-shot CoT | 238 | 0.727 | 0.667 | 0.780 | 4.485 | 9.543 |
| ARC-Challenge | Gemma-4-E4B | Few-shot CoT | 238 | 0.891 | 0.845 | 0.924 | 0.831 | 14.895 |
| ARC-Challenge | Phi-4-Mini-Reasoning | Few-shot CoT | 238 | 0.290 | 0.236 | 0.351 | 4.088 | 7.145 |
| ARC-Challenge | Phi-4-Reasoning | Few-shot CoT | 238 | 0.277 | 0.224 | 0.337 | 4.854 | 27.305 |
| ARC-Challenge | Qwen3-30B-A3B | Few-shot CoT | 238 | 0.487 | 0.425 | 0.551 | 9.651 | 57.621 |
| ARC-Challenge | Qwen3-8B | Few-shot CoT | 238 | 0.525 | 0.462 | 0.588 | 4.889 | 15.256 |

Continued on next page

Table 14: Full per-condition results under the matched 238-example core protocol (continued).

| Dataset | Model | Strategy | N | Accuracy | CI Low | CI High | Latency (s) | VRAM (GB) |
|---------|-------|----------|---|----------|--------|---------|-------------|-----------|
| ARC-Challenge | Gemma-4-26B-A4B | Zero-shot | 238 | 0.945 | 0.909 | 0.968 | 3.433 | 48.067 |
| ARC-Challenge | Gemma-4-E2B | Zero-shot | 238 | 0.769 | 0.711 | 0.818 | 1.379 | 9.543 |
| ARC-Challenge | Gemma-4-E4B | Zero-shot | 238 | 0.899 | 0.854 | 0.931 | 0.154 | 14.895 |
| ARC-Challenge | Phi-4-Mini-Reasoning | Zero-shot | 238 | 0.273 | 0.220 | 0.333 | 4.089 | 7.145 |
| ARC-Challenge | Phi-4-Reasoning | Zero-shot | 238 | 0.303 | 0.248 | 0.364 | 4.853 | 27.305 |
| ARC-Challenge | Qwen3-30B-A3B | Zero-shot | 238 | 0.416 | 0.355 | 0.479 | 9.605 | 57.621 |
| ARC-Challenge | Qwen3-8B | Zero-shot | 238 | 0.445 | 0.384 | 0.509 | 4.897 | 15.256 |
| GSM8K | Gemma-4-26B-A4B | CoT | 238 | 0.782 | 0.725 | 0.829 | 11.617 | 48.067 |
| GSM8K | Gemma-4-E2B | CoT | 238 | 0.685 | 0.623 | 0.741 | 5.015 | 9.543 |
| GSM8K | Gemma-4-E4B | CoT | 238 | 0.790 | 0.734 | 0.837 | 5.076 | 14.895 |
| GSM8K | Phi-4-Mini-Reasoning | CoT | 238 | 0.210 | 0.163 | 0.266 | 8.433 | 7.145 |
| GSM8K | Phi-4-Reasoning | CoT | 238 | 0.008 | 0.002 | 0.030 | 9.810 | 27.305 |
| GSM8K | Qwen3-30B-A3B | CoT | 238 | 0.756 | 0.698 | 0.806 | 14.415 | 57.621 |
| GSM8K | Qwen3-8B | CoT | 238 | 0.790 | 0.734 | 0.837 | 6.871 | 15.256 |
| GSM8K | Gemma-4-26B-A4B | Few-shot CoT | 238 | 0.681 | 0.619 | 0.737 | 14.346 | 48.067 |
| GSM8K | Gemma-4-E2B | Few-shot CoT | 238 | 0.710 | 0.649 | 0.764 | 4.515 | 9.543 |
| GSM8K | Gemma-4-E4B | Few-shot CoT | 238 | 0.752 | 0.694 | 0.803 | 5.405 | 14.895 |
| GSM8K | Phi-4-Mini-Reasoning | Few-shot CoT | 238 | 0.315 | 0.259 | 0.377 | 8.153 | 7.145 |
| GSM8K | Phi-4-Reasoning | Few-shot CoT | 238 | 0.021 | 0.009 | 0.048 | 9.660 | 27.305 |
| GSM8K | Qwen3-30B-A3B | Few-shot CoT | 238 | 0.807 | 0.752 | 0.852 | 14.740 | 57.621 |
| GSM8K | Qwen3-8B | Few-shot CoT | 238 | 0.819 | 0.765 | 0.863 | 6.674 | 15.256 |
| GSM8K | Gemma-4-26B-A4B | Zero-shot | 238 | 0.794 | 0.738 | 0.841 | 8.921 | 48.067 |
| GSM8K | Gemma-4-E2B | Zero-shot | 238 | 0.693 | 0.632 | 0.748 | 4.549 | 9.543 |
| GSM8K | Gemma-4-E4B | Zero-shot | 238 | 0.790 | 0.734 | 0.837 | 4.931 | 14.895 |

Table 14: Full per-condition results under the matched 238-example core protocol (continued).

| Dataset | | Model | Strategy | N | Accuracy | CI Low | CI High | Latency (s) | VRAM (GB) |
|---|---|---|---|---|---|---|---|---|---|
| GSM8K | | Phi-4-Mini-Reasoning | Zero-shot | 238 | 0.181 | 0.137 | 0.235 | 8.328 | 7.145 |
| GSM8K | | Phi-4-Reasoning | Zero-shot | 238 | 0.042 | 0.023 | 0.076 | 9.731 | 27.305 |
| GSM8K | | Qwen3-30B-A3B | Zero-shot | 238 | 0.782 | 0.725 | 0.829 | 14.302 | 57.621 |
| GSM8K | | Qwen3-8B | Zero-shot | 238 | 0.790 | 0.734 | 0.837 | 6.664 | 15.256 |
| MATH | L1–L3 | Gemma-4-26B-A4B | CoT | 238 | 0.643 | 0.580 | 0.701 | 17.412 | 48.067 |
| MATH | L1–L3 | Gemma-4-E2B | CoT | 238 | 0.580 | 0.516 | 0.641 | 10.964 | 9.543 |
| MATH | L1–L3 | Gemma-4-E4B | CoT | 238 | 0.668 | 0.606 | 0.725 | 10.833 | 14.895 |
| MATH | L1–L3 | Phi-4-Mini-Reasoning | CoT | 238 | 0.223 | 0.174 | 0.280 | 16.378 | 7.145 |
| MATH | L1–L3 | Phi-4-Reasoning | CoT | 238 | 0.013 | 0.004 | 0.036 | 19.318 | 27.305 |
| MATH | L1–L3 | Qwen3-30B-A3B | CoT | 238 | 0.576 | 0.512 | 0.637 | 26.157 | 57.621 |
| MATH | L1–L3 | Qwen3-8B | CoT | 238 | 0.277 | 0.224 | 0.337 | 13.506 | 15.256 |
| MATH | L1–L3 | Gemma-4-26B-A4B | Few-shot CoT | 238 | 0.571 | 0.508 | 0.633 | 23.528 | 48.067 |
| MATH | L1–L3 | Gemma-4-E2B | Few-shot CoT | 238 | 0.689 | 0.628 | 0.744 | 8.969 | 9.543 |
| MATH | L1–L3 | Gemma-4-E4B | Few-shot CoT | 238 | 0.693 | 0.632 | 0.748 | 8.278 | 14.895 |
| MATH | L1–L3 | Phi-4-Mini-Reasoning | Few-shot CoT | 238 | 0.391 | 0.331 | 0.454 | 16.543 | 7.145 |
| MATH | L1–L3 | Phi-4-Reasoning | Few-shot CoT | 238 | 0.029 | 0.014 | 0.059 | 19.389 | 27.305 |
| MATH | L1–L3 | Qwen3-30B-A3B | Few-shot CoT | 238 | 0.613 | 0.550 | 0.673 | 24.786 | 57.621 |
| MATH | L1–L3 | Qwen3-8B | Few-shot CoT | 238 | 0.639 | 0.576 | 0.697 | 11.515 | 15.256 |
| MATH | L1–L3 | Gemma-4-26B-A4B | Zero-shot | 238 | 0.693 | 0.632 | 0.748 | 14.125 | 48.067 |
| MATH | L1–L3 | Gemma-4-E2B | Zero-shot | 238 | 0.618 | 0.555 | 0.677 | 11.851 | 9.543 |
| MATH | L1–L3 | Gemma-4-E4B | Zero-shot | 238 | 0.630 | 0.567 | 0.689 | 11.952 | 14.895 |
| MATH | L1–L3 | Phi-4-Mini-Reasoning | Zero-shot | 238 | 0.252 | 0.201 | 0.311 | 16.289 | 7.145 |
| MATH | L1–L3 | Phi-4-Reasoning | Zero-shot | 238 | 0.038 | 0.020 | 0.070 | 19.265 | 27.305 |
| MATH | L1–L3 | Qwen3-30B-A3B | Zero-shot | 238 | 0.584 | 0.521 | 0.645 | 27.392 | 57.621 |

Table 14: Full per-condition results under the matched 238-example core protocol (continued).

| Dataset | Model | Strategy | N | Accuracy | CI Low | CI High | Latency (s) | VRAM (GB) |
|---|---|---|---|---|---|---|---|---|
| MATH L1–L3 | Qwen3-8B | Zero-shot | 238 | 0.218 | 0.171 | 0.275 | 13.270 | 15.256 |
| TruthfulQA MC1 | Gemma-4-26B-A4B | CoT | 238 | 0.727 | 0.667 | 0.780 | 2.698 | 48.067 |
| TruthfulQA MC1 | Gemma-4-E2B | CoT | 238 | 0.563 | 0.500 | 0.625 | 1.273 | 9.543 |
| TruthfulQA MC1 | Gemma-4-E4B | CoT | 238 | 0.643 | 0.580 | 0.701 | 2.476 | 14.895 |
| TruthfulQA MC1 | Phi-4-Mini-Reasoning | CoT | 238 | 0.983 | 0.958 | 0.993 | 4.092 | 7.145 |
| TruthfulQA MC1 | Phi-4-Reasoning | CoT | 238 | 0.996 | 0.977 | 0.999 | 4.600 | 27.305 |
| TruthfulQA MC1 | Qwen3-30B-A3B | CoT | 238 | 0.987 | 0.964 | 0.996 | 9.961 | 57.621 |
| TruthfulQA MC1 | Qwen3-8B | CoT | 238 | 0.987 | 0.964 | 0.996 | 5.156 | 15.256 |
| TruthfulQA MC1 | Gemma-4-26B-A4B | Few-shot CoT | 238 | 0.786 | 0.729 | 0.833 | 1.681 | 48.067 |
| TruthfulQA MC1 | Gemma-4-E2B | Few-shot CoT | 238 | 0.517 | 0.454 | 0.580 | 6.020 | 9.543 |
| TruthfulQA MC1 | Gemma-4-E4B | Few-shot CoT | 238 | 0.739 | 0.680 | 0.791 | 0.195 | 14.895 |
| TruthfulQA MC1 | Phi-4-Mini-Reasoning | Few-shot CoT | 238 | 0.966 | 0.935 | 0.983 | 4.143 | 7.145 |
| TruthfulQA MC1 | Phi-4-Reasoning | Few-shot CoT | 238 | 1.000 | 0.984 | 1.000 | 4.362 | 27.305 |
| TruthfulQA MC1 | Qwen3-30B-A3B | Few-shot CoT | 238 | 0.966 | 0.935 | 0.983 | 9.625 | 57.621 |
| TruthfulQA MC1 | Qwen3-8B | Few-shot CoT | 238 | 0.979 | 0.952 | 0.991 | 5.021 | 15.256 |
| TruthfulQA MC1 | Gemma-4-26B-A4B | Zero-shot | 238 | 0.794 | 0.738 | 0.841 | 2.653 | 48.067 |
| TruthfulQA MC1 | Gemma-4-E2B | Zero-shot | 238 | 0.550 | 0.487 | 0.612 | 2.613 | 9.543 |
| TruthfulQA MC1 | Gemma-4-E4B | Zero-shot | 238 | 0.735 | 0.676 | 0.787 | 0.456 | 14.895 |
| TruthfulQA MC1 | Phi-4-Mini-Reasoning | Zero-shot | 238 | 0.945 | 0.909 | 0.968 | 4.129 | 7.145 |
| TruthfulQA MC1 | Phi-4-Reasoning | Zero-shot | 238 | 0.996 | 0.977 | 0.999 | 4.403 | 27.305 |
| TruthfulQA MC1 | Qwen3-30B-A3B | Zero-shot | 238 | 0.983 | 0.958 | 0.993 | 9.907 | 57.621 |
| TruthfulQA MC1 | Qwen3-8B | Zero-shot | 238 | 0.962 | 0.930 | 0.980 | 4.586 | 15.256 |

## E  Larger-Sample Per-Condition Evaluation Matrix

This appendix reports the larger-sample per-condition results for ARC-Challenge, GSM8K, and TruthfulQA MC1, each using 500 examples per condition. MATH L1–L3 uses the same complete 238-example test subset

as the matched-core analysis, so its unchanged rows are not repeated. The table reports accuracy, Wilson confidence intervals, mean latency, and peak VRAM.

The TruthfulQA MC1 rows retain the original prepared answer ordering and are included only as protocol records. Substantive TruthfulQA interpretation and the principal weighted analyses use the corrected shuffled-choice results in Table 9.

Table 15: Larger-sample per-condition results for ARC-Challenge, GSM8K, and TruthfulQA MC1 at 500 examples per condition. The TruthfulQA MC1 rows retain the original prepared ordering and are included only as protocol records; substantive interpretation uses Table 9.

| Dataset | Model | Strategy | N | Acc. | 95% CI | Latency | VRAM |
|---|---|---|---|---|---|---|---|
| arc-challenge | Gemma-4-26B-A4B | cot | 500 | 0.858 | [0.825, 0.886] | 2.510 | 48.067 |
| arc-challenge | Gemma-4-E2B | cot | 500 | 0.766 | [0.727, 0.801] | 0.466 | 9.543 |
| arc-challenge | Gemma-4-E4B | cot | 500 | 0.878 | [0.846, 0.904] | 0.648 | 14.895 |
| arc-challenge | Phi-4-mini-reasoning | cot | 500 | 0.248 | [0.212, 0.288] | 4.051 | 7.145 |
| arc-challenge | Phi-4-reasoning | cot | 500 | 0.268 | [0.231, 0.308] | 4.859 | 27.305 |
| arc-challenge | Qwen3-30B-A3B | cot | 500 | 0.372 | [0.331, 0.415] | 9.539 | 57.674 |
| arc-challenge | Qwen3-8B | cot | 500 | 0.312 | [0.273, 0.354] | 5.160 | 15.256 |
| arc-challenge | Gemma-4-26B-A4B | few-shot-cot | 500 | 0.912 | [0.884, 0.934] | 2.572 | 48.067 |
| arc-challenge | Gemma-4-E2B | few-shot-cot | 500 | 0.730 | [0.689, 0.767] | 4.192 | 9.543 |
| arc-challenge | Gemma-4-E4B | few-shot-cot | 500 | 0.884 | [0.853, 0.909] | 0.711 | 14.895 |
| arc-challenge | Phi-4-mini-reasoning | few-shot-cot | 500 | 0.270 | [0.233, 0.311] | 4.063 | 7.145 |
| arc-challenge | Phi-4-reasoning | few-shot-cot | 500 | 0.246 | [0.210, 0.286] | 4.785 | 27.305 |
| arc-challenge | Qwen3-30B-A3B | few-shot-cot | 500 | 0.450 | [0.407, 0.494] | 9.227 | 57.671 |
| arc-challenge | Qwen3-8B | few-shot-cot | 500 | 0.508 | [0.464, 0.552] | 4.877 | 15.256 |
| arc-challenge | Gemma-4-26B-A4B | zero-shot | 500 | 0.922 | [0.895, 0.942] | 3.446 | 48.067 |
| arc-challenge | Gemma-4-E2B | zero-shot | 500 | 0.762 | [0.723, 0.797] | 1.629 | 9.543 |
| arc-challenge | Gemma-4-E4B | zero-shot | 500 | 0.906 | [0.877, 0.929] | 0.189 | 14.895 |
| arc-challenge | Phi-4-mini-reasoning | zero-shot | 500 | 0.248 | [0.212, 0.288] | 4.059 | 7.145 |
| arc-challenge | Phi-4-reasoning | zero-shot | 500 | 0.274 | [0.237, 0.315] | 4.833 | 27.305 |
| arc-challenge | Qwen3-30B-A3B | zero-shot | 500 | 0.384 | [0.342, 0.427] | 9.434 | 57.673 |
| arc-challenge | Qwen3-8B | zero-shot | 500 | 0.404 | [0.362, 0.448] | 4.723 | 15.256 |
| gsm8k | Gemma-4-26B-A4B | cot | 500 | 0.754 | [0.714, 0.790] | 11.469 | 48.067 |
| gsm8k | Gemma-4-E2B | cot | 500 | 0.656 | [0.613, 0.696] | 5.105 | 9.543 |
| gsm8k | Gemma-4-E4B | cot | 500 | 0.784 | [0.746, 0.818] | 5.171 | 14.895 |
| gsm8k | Phi-4-mini-reasoning | cot | 500 | 0.180 | [0.149, 0.216] | 8.195 | 7.145 |
| gsm8k | Phi-4-reasoning | cot | 500 | 0.020 | [0.011, 0.036] | 9.652 | 27.305 |
| gsm8k | Qwen3-30B-A3B | cot | 500 | 0.740 | [0.700, 0.777] | 14.188 | 57.673 |
| gsm8k | Qwen3-8B | cot | 500 | 0.776 | [0.737, 0.810] | 7.094 | 15.256 |
| gsm8k | Gemma-4-26B-A4B | few-shot-cot | 500 | 0.684 | [0.642, 0.723] | 13.866 | 48.067 |
| gsm8k | Gemma-4-E2B | few-shot-cot | 500 | 0.718 | [0.677, 0.756] | 4.685 | 9.543 |
| gsm8k | Gemma-4-E4B | few-shot-cot | 500 | 0.738 | [0.698, 0.775] | 5.671 | 14.895 |
| gsm8k | Phi-4-mini-reasoning | few-shot-cot | 500 | 0.316 | [0.277, 0.358] | 8.093 | 7.145 |
| gsm8k | Phi-4-reasoning | few-shot-cot | 500 | 0.018 | [0.009, 0.034] | 9.722 | 27.305 |
| gsm8k | Qwen3-30B-A3B | few-shot-cot | 500 | 0.762 | [0.723, 0.797] | 14.033 | 57.670 |
| gsm8k | Qwen3-8B | few-shot-cot | 500 | 0.812 | [0.775, 0.844] | 6.918 | 15.256 |
| gsm8k | Gemma-4-26B-A4B | zero-shot | 500 | 0.780 | [0.742, 0.814] | 8.297 | 48.067 |
| gsm8k | Gemma-4-E2B | zero-shot | 500 | 0.674 | [0.632, 0.714] | 4.760 | 9.543 |
| gsm8k | Gemma-4-E4B | zero-shot | 500 | 0.772 | [0.733, 0.807] | 4.985 | 14.895 |
| gsm8k | Phi-4-mini-reasoning | zero-shot | 500 | 0.150 | [0.121, 0.184] | 8.042 | 7.145 |
| gsm8k | Phi-4-reasoning | zero-shot | 500 | 0.032 | [0.020, 0.051] | 9.781 | 27.305 |
| gsm8k | Qwen3-30B-A3B | zero-shot | 500 | 0.752 | [0.712, 0.788] | 14.269 | 57.673 |
| gsm8k | Qwen3-8B | zero-shot | 500 | 0.770 | [0.731, 0.805] | 6.876 | 15.256 |

Table 15: Larger-sample per-condition results (continued).

| Dataset | Model | Strategy | N | Acc. | 95% CI | Latency | VRAM |
|---|---|---|---|---|---|---|---|
| truthfulqa-mc1 | Gemma-4-26B-A4B | cot | 500 | 0.750 | [0.710, 0.786] | 2.505 | 48.067 |
| truthfulqa-mc1 | Gemma-4-E2B | cot | 500 | 0.562 | [0.518, 0.605] | 1.224 | 9.543 |
| truthfulqa-mc1 | Gemma-4-E4B | cot | 500 | 0.628 | [0.585, 0.669] | 2.826 | 14.895 |
| truthfulqa-mc1 | Phi-4-mini-reasoning | cot | 500 | 0.984 | [0.969, 0.992] | 4.084 | 7.145 |
| truthfulqa-mc1 | Phi-4-reasoning | cot | 500 | 0.994 | [0.983, 0.998] | 4.910 | 27.305 |
| truthfulqa-mc1 | Qwen3-30B-A3B | cot | 500 | 0.984 | [0.969, 0.992] | 9.601 | 57.672 |
| truthfulqa-mc1 | Qwen3-8B | cot | 500 | 0.990 | [0.977, 0.996] | 5.192 | 15.256 |
| truthfulqa-mc1 | Gemma-4-26B-A4B | few-shot-cot | 500 | 0.820 | [0.784, 0.851] | 1.732 | 48.067 |
| truthfulqa-mc1 | Gemma-4-E2B | few-shot-cot | 500 | 0.518 | [0.474, 0.561] | 5.888 | 9.543 |
| truthfulqa-mc1 | Gemma-4-E4B | few-shot-cot | 500 | 0.742 | [0.702, 0.778] | 0.307 | 14.895 |
| truthfulqa-mc1 | Phi-4-mini-reasoning | few-shot-cot | 500 | 0.966 | [0.946, 0.979] | 4.128 | 7.145 |
| truthfulqa-mc1 | Phi-4-reasoning | few-shot-cot | 500 | 1.000 | [0.992, 1.000] | 4.800 | 27.305 |
| truthfulqa-mc1 | Qwen3-30B-A3B | few-shot-cot | 500 | 0.982 | [0.966, 0.991] | 9.671 | 57.672 |
| truthfulqa-mc1 | Qwen3-8B | few-shot-cot | 500 | 0.986 | [0.971, 0.993] | 5.153 | 15.256 |
| truthfulqa-mc1 | Gemma-4-26B-A4B | zero-shot | 500 | 0.792 | [0.754, 0.825] | 2.528 | 48.067 |
| truthfulqa-mc1 | Gemma-4-E2B | zero-shot | 500 | 0.546 | [0.502, 0.589] | 2.595 | 9.543 |
| truthfulqa-mc1 | Gemma-4-E4B | zero-shot | 500 | 0.732 | [0.692, 0.769] | 0.545 | 14.895 |
| truthfulqa-mc1 | Phi-4-mini-reasoning | zero-shot | 500 | 0.954 | [0.932, 0.969] | 4.100 | 7.145 |
| truthfulqa-mc1 | Phi-4-reasoning | zero-shot | 500 | 0.996 | [0.986, 0.999] | 4.884 | 27.305 |
| truthfulqa-mc1 | Qwen3-30B-A3B | zero-shot | 500 | 0.990 | [0.977, 0.996] | 9.674 | 57.671 |
| truthfulqa-mc1 | Qwen3-8B | zero-shot | 500 | 0.962 | [0.941, 0.976] | 4.666 | 15.256 |

# F   Additional Statistical Results

This appendix reports supplemental statistical tables supporting the main deployment-aware interpretation. Table 16 reports the corrected TruthfulQA MC1 matched-subset and larger-matrix results used in the principal weighted analyses. Table 17 reports bootstrap confidence intervals for the strongest weighted configurations. Table 18 reports paired permutation comparisons among the strongest weighted configurations with Holm correction. Table 19 reports a compact model-level summary of weighted rank instability across prompting strategies. Table 20 reports budget-constrained and efficiency-oriented operating points. Table 21 reports compact Phi-4-Reasoning compatibility diagnostics. Table 22 reports weight-sensitivity winners. Table 23 reports the direct precision, load-mode, dtype, and quantization-state audit.

Table 16: TruthfulQA MC1 results for both the matched 238-example subset and the larger 500-example shuffled-choice matrix. Each cell reports accuracy with the Wilson 95% confidence interval in brackets and the missing-prediction rate in parentheses. Missing-prediction rate is the fraction of examples for which the strict extractor did not return a scoreable prediction.

| Model | Zero-shot | CoT | Few-shot CoT |
|---|---|---|---|
| **Matched 238-example subset** | | | |
| Gemma-4-E2B | 0.492 [0.429, 0.555] (0.004) | 0.517 [0.454, 0.580] (0.004) | 0.521 [0.458, 0.584] (0.000) |
| Gemma-4-E4B | 0.664 [0.602, 0.721] (0.126) | 0.471 [0.408, 0.534] (0.433) | 0.714 [0.654, 0.768] (0.042) |
| Gemma-4-26B-A4B | 0.618 [0.555, 0.677] (0.282) | 0.403 [0.343, 0.467] (0.508) | 0.681 [0.619, 0.737] (0.155) |
| Phi-4-mini-reasoning | 0.008 [0.002, 0.030] (0.992) | 0.008 [0.002, 0.030] (0.992) | 0.004 [0.001, 0.023] (0.996) |
| Phi-4-reasoning | 0.097 [0.065, 0.141] (0.895) | 0.059 [0.035, 0.096] (0.937) | 0.008 [0.002, 0.030] (0.983) |
| Qwen3-30B-A3B | 0.126 [0.090, 0.174] (0.870) | 0.130 [0.093, 0.179] (0.861) | 0.193 [0.148, 0.248] (0.794) |
| Qwen3-8B | 0.109 [0.076, 0.155] (0.878) | 0.071 [0.045, 0.111] (0.924) | 0.139 [0.100, 0.188] (0.849) |
| **Larger 500-example matrix** | | | |
| Gemma-4-E2B | 0.500 [0.456, 0.544] (0.002) | 0.510 [0.466, 0.554] (0.002) | 0.530 [0.486, 0.573] (0.000) |
| Gemma-4-E4B | 0.604 [0.560, 0.646] (0.162) | 0.422 [0.379, 0.466] (0.476) | 0.690 [0.648, 0.729] (0.050) |
| Gemma-4-26B-A4B | 0.602 [0.558, 0.644] (0.298) | 0.418 [0.376, 0.462] (0.508) | 0.698 [0.656, 0.737] (0.156) |
| Phi-4-mini-reasoning | 0.008 [0.003, 0.020] (0.984) | 0.004 [0.001, 0.014] (0.996) | 0.004 [0.001, 0.014] (0.994) |
| Phi-4-reasoning | 0.100 [0.077, 0.129] (0.888) | 0.054 [0.037, 0.077] (0.942) | 0.006 [0.002, 0.017] (0.976) |
| Qwen3-30B-A3B | 0.136 [0.109, 0.169] (0.854) | 0.122 [0.096, 0.154] (0.868) | 0.202 [0.169, 0.239] (0.786) |
| Qwen3-8B | 0.100 [0.077, 0.129] (0.888) | 0.058 [0.041, 0.082] (0.940) | 0.124 [0.098, 0.156] (0.866) |

Table 17: Bootstrap summary for the six strongest weighted configurations under the matched-core protocol. Weighted scores use the stated task weights and corrected deterministic option-count-aware TruthfulQA MC1 rows. Intervals are percentile intervals from 5000 bootstrap resamples.

| Model | Strategy | Point estimate | Bootstrap mean | 95% CI low | 95% CI high | Bootstrap std. |
|---|---|---|---|---|---|---|
| Gemma-4-26B-A4B | Zero-shot | 0.776 | 0.777 | 0.748 | 0.804 | 0.0147 |
| Gemma-4-E4B | Few-shot CoT | 0.758 | 0.758 | 0.728 | 0.788 | 0.0152 |
| Gemma-4-E4B | Zero-shot | 0.751 | 0.751 | 0.722 | 0.779 | 0.0149 |
| Gemma-4-E4B | CoT | 0.742 | 0.742 | 0.712 | 0.770 | 0.0149 |
| Gemma-4-26B-A4B | CoT | 0.724 | 0.724 | 0.695 | 0.753 | 0.0150 |
| Gemma-4-26B-A4B | Few-shot CoT | 0.699 | 0.699 | 0.668 | 0.731 | 0.0162 |

Table 18: Paired permutation comparisons with Holm correction among top weighted configurations. Raw two-sided p-values and Holm-adjusted p-values are reported to avoid overinterpreting multiple top-row comparisons.

| Model A / Strategy A | Model B / Strategy B | Delta | Raw $p$ | Holm adj. $p$ |
|---|---|---|---|---|
| Gemma-4-26B-A4B / Zero-shot | Gemma-4-E4B / Few-shot CoT | 0.018 | 0.2132 | 0.8528 |
| Gemma-4-26B-A4B / Zero-shot | Gemma-4-E4B / Zero-shot | 0.025 | 0.0875 | 0.4377 |
| Gemma-4-26B-A4B / Zero-shot | Gemma-4-E4B / CoT | 0.035 | 0.0177 | 0.1065 |
| Gemma-4-26B-A4B / Zero-shot | Gemma-4-26B-A4B / CoT | 0.053 | 0.0000 | 0.0003 |
| Gemma-4-E4B / Few-shot CoT | Gemma-4-E4B / Zero-shot | 0.007 | 0.6090 | 0.8528 |
| Gemma-4-E4B / Few-shot CoT | Gemma-4-E4B / CoT | 0.017 | 0.2291 | 0.8528 |
| Gemma-4-E4B / Zero-shot | Gemma-4-E4B / CoT | 0.010 | 0.3933 | 0.8528 |

Table 19: Compact prompt-instability summary across weighted rankings. Weighted rank range is the difference between the best and worst weighted rank of a model across CoT, few-shot CoT, and zero-shot prompting. Dataset best-strategy flips indicates whether the model's best prompting strategy changes across the four benchmark datasets.

| Model | Rank (CoT) | Rank (Few-shot CoT) | Rank (Zero-shot) | Weighted rank range | Dataset best-strategy flips |
|---|---|---|---|---|---|
| Gemma-4-E4B | 1 | 1 | 2 | 1 | 1 |
| Gemma-4-26B-A4B | 2 | 2 | 1 | 1 | 1 |
| Gemma-4-E2B | 3 | 3 | 3 | 0 | 1 |
| Qwen3-8B | 5 | 4 | 5 | 1 | 1 |
| Qwen3-30B-A3B | 4 | 5 | 4 | 1 | 0 |
| Phi-4-Mini-Reasoning | 6 | 6 | 6 | 0 | 1 |
| Phi-4-Reasoning | 7 | 7 | 7 | 0 | 0 |

Table 20: Deployment-budget and efficiency summary under the matched-core protocol. The upper block reports the best weighted configuration under selected VRAM budgets. The lower block reports the strongest configurations by the combined efficiency score weighted accuracy divided by latency and VRAM.

| Summary type | Model | Strategy | Weighted acc. | Latency (s) | VRAM (GB) |
|---|---|---|---|---|---|
| Best under ≤16 GB | Gemma-4-E4B | Few-shot CoT | 0.758 | 3.772 | 14.895 |
| Best under ≤24 GB | Gemma-4-E4B | Few-shot CoT | 0.758 | 3.772 | 14.895 |
| Best under ≤48 GB | Gemma-4-E4B | Few-shot CoT | 0.758 | 3.772 | 14.895 |
| Best unrestricted | Gemma-4-26B-A4B | Zero-shot | 0.776 | 7.432 | 48.067 |
| Top efficiency score | Gemma-4-E2B | CoT | 0.653 | 4.527 | 9.543 |
| Second efficiency score | Gemma-4-E4B | Few-shot CoT | 0.758 | 3.772 | 14.895 |
| Third efficiency score | Gemma-4-E2B | Zero-shot | 0.666 | 5.476 | 9.543 |

Table 21: Compatibility diagnostics for Phi-4-Reasoning under the unified evaluation pipeline. Missing-prediction rate corresponds to the share of cases with no scoreable extracted prediction. Malformed-output rate is reported for math-style tasks where the expected final-answer format is explicit; N/A indicates that this diagnostic is not applicable to that task format. The TruthfulQA MC1 rows use the original prepared answer ordering and are retained only as protocol diagnostics; substantive TruthfulQA interpretation should use the shuffled-choice audit in Table 9.

| Dataset | Strategy | Accuracy | Missing pred. rate | Think tag rate | Malformed output rate |
|---|---|---|---|---|---|
| ARC-Challenge | CoT | 0.294 | 0.000 | 0.992 | N/A |
| ARC-Challenge | Few-shot CoT | 0.277 | 0.000 | 1.000 | N/A |
| ARC-Challenge | Zero-shot | 0.303 | 0.000 | 0.950 | N/A |
| GSM8K | CoT | 0.008 | 0.983 | 1.000 | 0.139 |
| GSM8K | Few-shot CoT | 0.021 | 0.975 | 0.992 | 0.059 |
| GSM8K | Zero-shot | 0.042 | 0.958 | 0.975 | 0.155 |
| MATH L1–L3 | CoT | 0.013 | 0.000 | 0.962 | 0.122 |
| MATH L1–L3 | Few-shot CoT | 0.029 | 0.000 | 0.954 | 0.101 |
| MATH L1–L3 | Zero-shot | 0.038 | 0.000 | 0.924 | 0.122 |
| TruthfulQA MC1 | CoT | 0.996 | 0.000 | 0.950 | N/A |
| TruthfulQA MC1 | Few-shot CoT | 1.000 | 0.000 | 1.000 | N/A |
| TruthfulQA MC1 | Zero-shot | 0.996 | 0.000 | 0.916 | N/A |

Table 22: Weight-sensitivity winners under matched-core and larger-sample matrices. Each row reports the top configuration under the corresponding task-weight scheme. All TruthfulQA components use corrected deterministic option-count-aware shuffled-choice results rather than original-order protocol rows.

| Matrix | Weight scheme | Top configuration | Weighted score |
|---|---|---|---|
| Matched core | Stated weighting | Gemma-4-26B-A4B / Zero-shot | 0.776 |
| Matched core | Equal | Gemma-4-E4B / Few-shot CoT | 0.763 |
| Matched core | ARC-heavy | Gemma-4-26B-A4B / Zero-shot | 0.832 |
| Matched core | GSM8K-heavy | Gemma-4-26B-A4B / Zero-shot | 0.761 |
| Matched core | MATH-heavy | Gemma-4-26B-A4B / Zero-shot | 0.741 |
| Matched core | TruthfulQA-heavy | Gemma-4-E4B / Few-shot CoT | 0.735 |
| Larger-sample | Stated weighting | Gemma-4-26B-A4B / Zero-shot | 0.765 |
| Larger-sample | Equal | Gemma-4-E4B / Few-shot CoT | 0.751 |
| Larger-sample | ARC-heavy | Gemma-4-26B-A4B / Zero-shot | 0.816 |
| Larger-sample | GSM8K-heavy | Gemma-4-26B-A4B / Zero-shot | 0.750 |
| Larger-sample | MATH-heavy | Gemma-4-26B-A4B / Zero-shot | 0.733 |
| Larger-sample | TruthfulQA-heavy | Gemma-4-E4B / Few-shot CoT | 0.720 |

Table 23: Direct bf16 load audit for the seven evaluated model configurations on an H100 GPU. All models reported no 4-bit or 8-bit loading and no quantization configuration. Memory footprint is reported in GiB.

| Model | Dtype | 4-bit | 8-bit | Quant. config | Footprint (GiB) |
|---|---|---|---|---|---|
| Gemma-4-E2B | bf16 | No | No | None | 9.51 |
| Gemma-4-E4B | bf16 | No | No | None | 14.79 |
| Gemma-4-26B-A4B | bf16 | No | No | None | 48.06 |
| Qwen3-8B | bf16 | No | No | None | 15.26 |
| Qwen3-30B-A3B | bf16 | No | No | None | 56.86 |
| Phi-4-Mini-Reasoning | bf16 | No | No | None | 7.15 |
| Phi-4-Reasoning | bf16 | No | No | None | 27.31 |

The audit found zero `Linear4bit` or bitsandbytes modules and bfloat16 as the only parameter dtype. The audit environment did not include `bitsandbytes`. These results replace the earlier indirect footprint-ratio argument with direct load-state evidence.

