# OpenReview forum: "Unified Deployment-Aware Evaluation of Open Reasoning Language Models"
_TMLR — Under review for TMLR_

### Review · Reviewer_j7bb · 2026-07-02

**Summary Of Contributions:**

The paper runs a unified "deployment-aware" evaluation of seven open reasoning LLMs (Gemma-4-26B-A4B, Gemma-4-E2B, Gemma-4-E4B, Phi-4-Mini-Reasoning, Phi-4-Reasoning, Qwen3-30B-A3B, Qwen3-8B) across four benchmarks and three prompting strategies, all on a matched n=238 subset (84 conditions total). Beyond accuracy, they report latency, VRAM, a weighted aggregate score, Pareto analysis, an oracle routing upper bound, prompt-sensitivity statistics, and compatibility diagnostics. The main claim: the top weighted-score model (Gemma-4-26B-A4B, zero-shot) isn't the best deployment choice, since Gemma-4-E4B scores nearly as well at a fraction of the latency and memory cost. They also show prompting strategy reorders rankings rather than shifting everyone equally, and that Phi-4-Reasoning's poor math scores trace back to extraction failures rather than pure reasoning weakness.

**Audience:**

Yes

**Audience Explanation:**

Deployment-aware evaluation of open LLMs is timely, and the core message (leaderboard-best and deployment-best configurations diverge, and prompting is a real experimental factor rather than a footnote) is useful on its own terms. The rank-correlation analysis of prompting effects and the oracle-routing framing should interest practitioners doing model selection under resource constraints, and the "explain the failure, don't just report it" approach to the Phi-4-Reasoning results is a good template for other evaluation papers. The specific models will age quickly, but the methodology should hold up.

**Broader Impact Concerns:**

No major concerns.

**Claims And Evidence:**

No

**Claims Explanation:**

The statistical backbone is solid, but two issues weaken the central claims:

(1) Quantization is unresolved. Section 3.4 says the loader tries bf16 then falls back to 4-bit "when needed," but the authors admit they can't confirm bitsandbytes was actually working. Since precision affects both accuracy and the VRAM/latency numbers that drive the entire deployment-aware argument (Table 3, Figures 4-5, the Pareto frontier, the budget analysis), this is a real problem. If some conditions ran 4-bit and others bf16, the "Gemma-4-E4B is the efficient sweet spot" conclusion could just be an artifact of inconsistent precision rather than a real tradeoff.

(2) Extraction pipeline may be conflated with model capability. The Phi-4-Reasoning diagnostics are a highlight, but the answer-format rules are strict (exact terminal format required, no explanation allowed), and the dominant failure mode is near-universal think-tag usage. It's not clear whether the extraction code strips <think> spans before parsing. If it doesn't, "not robust enough for deployment" may really mean "the extraction script needs a fix," which is a different and much smaller claim.

There's also a minor circularity in the n=238 justification: the other three benchmarks have much larger native test sets and were downsampled just to match the filtered MATH L1-3 subset. This produces fairly wide CIs on individual conditions, so some of the fine-grained per-dataset comparisons (down to the third decimal) are less solid than the prose implies, even though the weighted-score-level claims are handled more carefully.

**Requested Changes:**

(1) Pin down the precision issue from Section 3.4. Confirm per-model, per-condition whether bf16 or 4-bit was actually used (e.g., check observed VRAM against known bf16 footprint, or re-run with bitsandbytes confirmed working). As it stands, the VRAM/latency numbers can't fully be trusted, and they're central to the paper's main claim.

(2) Clarify and, if needed, fix how the extraction pipeline handles <think> traces. Report whether results change under a more lenient parse (searching the full output, or stripping reasoning traces first). This matters a lot for how strongly the paper can claim Phi-4-Reasoning is deployment-unready, which is one of its five stated contributions.

(3) Justify n=238 more thoroughly, or increase it. ARC-Challenge, GSM8K, and TruthfulQA MC1 all have much larger test sets available, so capping everything to the smallest common subset needs more explanation. At minimum, flag which fine-grained claims (especially the third-decimal comparisons in Section 4.3) actually fall within the reported CIs.

---

> ### Author Response · Authors · 2026-07-16
> **Response to Reviewer j7bb (Part 1 of 2)**
>
> # Response to Reviewer j7bb
>
> We sincerely thank Reviewer j7bb for the careful and constructive review. We have uploaded a revised manuscript in which all changes are marked with yellow highlighted text. The revision addresses the precision/quantization ambiguity through a direct load-state audit, separates extraction-pipeline behavior from model capability through parser-aware diagnostics, and strengthens the n=238 justification through a 500-example robustness matrix and a 1000-draw repeated-subset analysis, with all claims scoped to the tested models, datasets, prompts, H100 batch-size-one bf16 setting, and extraction pipeline. During revision, we also corrected an answer-ordering issue in the prepared TruthfulQA MC1 records and updated the affected evaluations and analyses throughout the manuscript. A detailed point-by-point response is provided below.
>
> ---
>
> **Comment 1.** "Pin down the precision issue from Section 3.4. Confirm per-model, per-condition whether bf16 or 4-bit was actually used (e.g., check observed VRAM against known bf16 footprint, or re-run with bitsandbytes confirmed working). As it stands, the VRAM/latency numbers can't fully be trusted, and they're central to the paper's main claim."
>
> **Response.** We agree, and we addressed this with the strongest direct evidence recoverable from the retained artifacts and a follow-up runtime audit. The revised manuscript adds a load-only audit for all seven model configurations using the same model identifiers and bf16 loading path: the audit records `is_loaded_in_4bit=false`, `is_loaded_in_8bit=false`, `quantization_config=None`, zero `Linear4bit` modules, zero bitsandbytes modules, and bfloat16 parameter tensors for every audited model, with audited load footprints consistent with the observed per-model VRAM values in the main tables. We also now distinguish the retained main-run artifacts and software snapshot from the follow-up audit environment, and we word the conclusion conservatively: the retained load-mode fields, observed VRAM values, and direct load audit support interpreting the reported runs as H100, batch-size-one, bf16 executions under the reported stack, not as hardware-invariant costs. The released artifacts and audit protocol additionally allow the deployment measurements to be reproduced and extended on other hardware and software environments, which we identify as future work.
>
> **Manuscript location:** Section 3.4; Appendix C, Tables 12–13; Appendix F, Table 23; Section 5 (Limitations).
>
> ---
>
> **Comment 2.** "Clarify and, if needed, fix how the extraction pipeline handles <think> traces. Report whether results change under a more lenient parse (searching the full output, or stripping reasoning traces first). This matters a lot for how strongly the paper can claim Phi-4-Reasoning is deployment-unready, which is one of its five stated contributions."
>
> **Response.** The revision separates strict benchmark scoring from diagnostic parsing and documents both. Strict scoring applies the task-specific extractor to the raw decoded response and is used in all main tables; the diagnostic lenient parser strips trace-like spans such as `<think>...</think>` and searches the remaining text for likely answer content, and it is used only for failure attribution, never as a replacement benchmark score. For Phi-4-Reasoning we now report, per condition, strict accuracy, diagnostic lenient accuracy, missing-prediction rate, and the dominant diagnostic pattern — on GSM8K, missing-prediction rates are 0.958–0.983 with near-universal think-tag traces, while lenient diagnostic accuracy recovers substantially more likely answer content (e.g., 0.445–0.508 on GSM8K). We accordingly softened the contribution claim from model incapability to poor scoreability and interface adherence under the standardized extraction interface, which is the deployment-relevant finding.
>
> **Manuscript location:** Section 4.7; Table 7; Appendix B; Appendix F, Table 21.
>
> ---
> Continued in Part 2 of 2.

---

> ### Author Response · Authors · 2026-07-16
> **Response to Reviewer j7bb (Part 2 of 2)**
>
> **Comment 3.** "Justify n=238 more thoroughly, or increase it. ARC-Challenge, GSM8K, and TruthfulQA MC1 all have much larger test sets available, so capping everything to the smallest common subset needs more explanation. At minimum, flag which fine-grained claims (especially the third-decimal comparisons in Section 4.3) actually fall within the reported CIs."
>
> **Response.** We did both. First, MATH L1–L3 is now precisely defined as the complete level-1–3 portion of the configured MATH test split (43 Level 1, 90 Level 2, 105 Level 3 examples, 238 total), so 238 is the full available test subset for that component rather than an arbitrary cap; expanding it would require Level 4–5 or training-split examples, changing the benchmark definition. Second, we increased the sample for the other benchmarks: ARC-Challenge, GSM8K, and TruthfulQA MC1 are additionally evaluated at 500 examples per condition, and a 1000-draw repeated 238-example subset analysis reports mean, SD, 2.5th/97.5th percentiles, and winner frequencies (leader in 926/1000 draws; mean rank correlation 0.985 with the matched core). Third, the prose no longer over-reads third-decimal gaps: bootstrap intervals overlap, and Holm-adjusted paired permutation tests (e.g., leader versus Gemma-4-E4B few-shot CoT: raw p = 0.2132, Holm p = 0.8528) are now cited explicitly to flag which top-row differences should not be treated as decisive.
>
> **Manuscript location:** Section 3.1; Sections 4.2–4.3; Tables 2–4; Appendix E, Table 15; Appendix F, Tables 17–18.

---

### Review · Reviewer_wKiv · 2026-07-02

**Summary Of Contributions:**

This work presents an unified evaluation of seven open reasoning language models configurations across four benchmarks under three prompting strategies. It shows the distinction between the highest weighted-score configuration and the strongest practical operating point through Pareto-frontier analysis, deployment budget summaries, and resource-normalized efficiency metrics, and argues for a shift in evaluation style.

Strength:
1) Comprehensive experiments are conducted and the analysis and interpretation are helpful on understanding the problems and conclusions.
2) The variables are carefully controlled and analyzed, making the results and interpretation trustworthy.

Weakness and questions:
1) The selection on the language models, benchmarks and prompting strategies are unclear to me. If possible, please provide the rationale or references for the selection of these representative models and discuss whether they adequately represent the field. Additionally, please elaborate on the generalizability of the proposed framework and how it can be applied to other models, benchmarks, and prompting strategies.
2) While the sample subsets are controlled to be as large as possible across all benchmarks, is it possible to show the full-size performance as well, or do the cross validation so as for the robustness. And then please also report the variance in the Table.
3) Section 3.4 appears to be a little redundant to me. I suggest keeping it concise and emphasize more on the strategies and the rationale. Some details can be moved to the appendix.

**Additional Comments:**

NA

**Audience:**

Yes

**Audience Explanation:**

1) The comprehensive analysis helps to interpret the behavior of the tested models and provides guidelines for model comparison and selection.
2) The proposed frame is inspiring on future evaluation or benchmarking works.

**Claims And Evidence:**

Yes

**Claims Explanation:**

This work proposed a new and comprehensive view on evaluation of open reasoning language models. Comprehensive experiments with careful set up are conducted with rational analysis, which shows the distinction between the highest weighted-score configuration and the strongest practical operating point.

**Requested Changes:**

1) Rewrite the modules mentioned above with more details on the rationale for the selection or the models, the benchmarks and the prompting strategies in this work.
2) Add discussion on the generalizability of the framework applied in this work.
3) Full-size evaluation or cross-validation over the selected benchmarks.

---

> ### Author Response · Authors · 2026-07-16
> **Response to Reviewer wKiv (Part 1 of 2)**
>
> # Response to Reviewer wKiv
>
> We sincerely thank Reviewer wKiv for the positive and constructive feedback. We have uploaded a revised manuscript in which all changes are marked with yellow highlighted text. The revision adds explicit rationale for the model, benchmark, and prompting choices; a dedicated generalizability/extensibility discussion; a 500-example larger-sample robustness matrix and a 1000-draw repeated-subset stability analysis reporting variance estimates; and a reorganized Section 3.4 with detailed reproducibility material moved to the appendices. During revision, we also corrected an answer-ordering issue in the prepared TruthfulQA MC1 records and updated the affected evaluations and analyses throughout the manuscript. A detailed point-by-point response is provided below.
>
> ---
>
> **Comment 1.** "Rewrite the modules mentioned above with more details on the rationale for the selection or the models, the benchmarks and the prompting strategies in this work."
>
> **Response.** We added explicit rationale for all three choices. For benchmarks, the revision states that ARC-Challenge, GSM8K, MATH L1–L3, and TruthfulQA MC1 were selected to span complementary deployment-relevant failure modes — multiple-choice science reasoning, arithmetic word-problem reasoning, filtered mathematical problem solving, and truthfulness under plausible distractors — rather than to exhaustively represent all reasoning workloads. For models, the revision clarifies that the seven configurations were chosen to cover practically relevant open-model regimes: dense and mixture-of-experts architectures, smaller and larger active-parameter footprints, and three prominent open model families with reasoning-oriented releases, with the conclusions explicitly limited to these families and release interfaces. For prompting, the revision explains that zero-shot, CoT, and few-shot CoT were selected as common prompt families with different inference costs and output-format risks, standardized for sensitivity measurement rather than optimized for any benchmark.
>
> **Manuscript location:** Sections 3.1–3.3; Section 2.4.
>
> ---
>
> **Comment 2.** "Add discussion on the generalizability of the framework applied in this work."
>
> **Response.** We added a dedicated extensibility discussion. The revised manuscript states that extending the protocol to another model requires recording its model identifier, revision evidence, tokenizer and chat-template behavior, generation configuration, and load-state audit, and that extending it to another benchmark or prompting strategy requires a standardized example schema, selected example identifiers, a versioned prompt definition, a task-specific strict extraction rule, and rerunning deployment measurements in the target hardware and software environment. The released artifacts and audit protocol make the deployment measurements reproducible and extensible to other environments, which we identify as future work, while the empirical conclusions themselves are explicitly limited to the evaluated models, benchmarks, prompts, hardware, precision mode, and extraction pipeline.
>
> **Manuscript location:** Section 2.4; Section 3.4; Section 5 (Discussion, Limitations, and Future Work).
>
> ---
> Continued in Part 2 of 2.

---

> ### Author Response · Authors · 2026-07-16
> **Response to Reviewer wKiv (Part 2 of 2)**
>
> **Comment 3.** "Full-size evaluation or cross-validation over the selected benchmarks. ... please also report the variance in the Table."
>
> **Response.** We added a larger-sample robustness layer and a repeated-subset stability analysis with variance reported in the tables. ARC-Challenge, GSM8K, and TruthfulQA MC1 are now additionally evaluated at 500 examples per condition, while MATH L1–L3 uses all 238 level-1–3 examples in the configured MATH test split (43 Level 1, 90 Level 2, 105 Level 3); expanding MATH to 500 would require Level 4–5 or training-split examples and would change the benchmark definition. We further sampled 1000 repeated 238-example subsets from the larger non-MATH matrices with the complete MATH component fixed: Table 4 reports the mean, SD, 2.5th and 97.5th percentiles, and winner frequency for each top configuration (Gemma-4-26B-A4B zero-shot wins 926/1000 subsets, Gemma-4-E4B few-shot CoT 72/1000, Gemma-4-E4B zero-shot 2/1000; mean rank correlation with the matched core is 0.985, SD 0.004). Per-condition Wilson confidence intervals are reported throughout, including for the larger-sample matrix and the corrected TruthfulQA results.
>
> **Manuscript location:** Section 3.1; Section 4.2; Tables 2–4; Appendix E, Table 15; Appendix F, Tables 16–17.
>
> ---
>
> **Comment 4.** "Section 3.4 appears to be a little redundant to me. I suggest keeping it concise and emphasize more on the strategies and the rationale. Some details can be moved to the appendix."
>
> **Response.** We reorganized Section 3.4 and reduced duplication by moving the detailed prompt, extraction, dataset-source, model-revision, software-environment, and load-audit material to the appendices. The main text now emphasizes the inference protocol and the distinctions needed for interpretation: the retained main-run environment versus the follow-up audit environment, deterministic decoding, bf16/load-mode interpretation, and the environment dependence of latency and VRAM.
>
> **Manuscript location:** Section 3.4; Appendices A–C and F; Tables 11–13 and 23.

---

### Review · Reviewer_8gp8 · 2026-07-03

**Summary Of Contributions:**

The paper presents a unified empirical evaluation of seven open reasoning language model configurations across four benchmarks (ARC-Challenge, GSM8K, MATH L1–L3, and TruthfulQA MC1) under three prompting strategies: zero-shot, CoT, and few-shot CoT. All conditions are evaluated on the same 238-example subset, and the authors report accuracy, confidence intervals, latency, VRAM usage, weighted aggregate scores, Pareto-style tradeoffs, prompt sensitivity, and compatibility diagnostics.

The main claim is that open model evaluation should be framed as a deployment-aware operating-point problem rather than as a single-score leaderboard exercise. This is a reasonable and timely motivation. The complete model–dataset–prompt matrix and the inclusion of latency, memory, and output-format diagnostics are useful.

However, the contribution remains close to a benchmark report. The 238-example design is weakly justified, the weighted score is arbitrary, the prompting protocol is not always cleanly interpretable, and some failures (especially for Phi-4-Reasoning) appear to reflect parser or interface incompatibility rather than model capability. Overall, the message is plausible, but the evidence is too narrow and fragile.

**Audience:**

Yes

**Audience Explanation:**

Some TMLR readers would likely be interested in the findings, especially those working on LLM evaluation, open model benchmarking, prompt sensitivity, and practical model deployment. The paper addresses a timely and relevant issue: benchmark accuracy alone is not enough for model selection, and deployment constraints such as latency, memory, prompt robustness, and output-format adherence matter.

The empirical comparison may also be useful to practitioners who want a compact view of how several open reasoning-oriented models behave under a common pipeline. The observation that a smaller or more efficient model can be a more attractive operating point than the highest-scoring model is relevant to applied ML audiences. The diagnostic discussion around extraction failures and interface adherence is also potentially valuable.

That said, interest is not the same as publishability. The current version reads more like an internal benchmark report or a systems note than a mature TMLR research article. The topic is relevant to TMLR, but the present evidence and methodological contribution are not yet strong enough.

**Broader Impact Concerns:**

I do not see major ethical concerns that would by themselves require rejection. However, I do think the paper should probably include a short broader impact discussion.

**Claims And Evidence:**

No

**Claims Explanation:**

The paper supports some narrow descriptive claims about this particular evaluation pipeline, but not the broader claims made in the manuscript. The authors do show that, under their setup, models differ in accuracy, latency, memory use, prompt sensitivity, and output-format robustness. This is useful.

However, the evidence is not convincing enough for the stronger conclusions. The evaluation uses only one 238-example subset, selected with one seed, rather than full benchmark splits or repeated subsampling. The weighted aggregate score is arbitrary and nevertheless drives much of the narrative. The prompting protocol is also problematic: for multiple-choice tasks, CoT-style prompting is combined with rules asking the model to return only one letter and no explanation.

The Phi-4-Reasoning results are particularly concerning. Very high missing-prediction rates suggest a mismatch between the model output format and the extraction pipeline. Such failures are deployment-relevant, but they should probably be separated from wrong answers and not treated as clean evidence of reasoning performance.

Finally, the deployment analysis is mostly descriptive and depends on one hardware setup, batch size, decoding regime, and unclear precision/quantization behavior. Thus, the paper provides useful observations, but the claims are overstated relative to the evidence.

**Requested Changes:**

1) The authors should either evaluate on the full benchmark splits or provide a convincing stability analysis over multiple random subsets. A single 238-example subset with one seed is not enough to support the paper’s claims.

2) The Phi-4-Reasoning results should be rerun with a more appropriate extraction protocol, or at least reported with parser failures clearly separated from incorrect answers. Treating extraction failures as ordinary benchmark failures gives a misleading impression of model capability.

3) The authors should explain how CoT prompting is meaningful when the appended task rule says “return only one capital letter” and “do not output any explanation.” They should either redesign the prompts or substantially weaken the claims about prompt sensitivity.

4) The paper should specify exact model revisions, tokenizer versions, selected example IDs, prompt files, extraction rules, raw outputs or representative failures, precision modes, quantization behavior, and software versions. In a deployment-aware paper, ambiguity about bf16 versus 4-bit execution is a serious issue.

5) The weighted score should either be justified through a deployment utility model or treated as only one illustrative summary. The paper should avoid making the weighted ranking the central evidence unless the weights are meaningfully motivated.

6) The authors should explain the bootstrap and permutation procedures more carefully, including the resampling unit and dependence structure. They should also address multiple comparisons and show robustness across sampled subsets.

7) The paper should make clearer what is new relative to existing LLM evaluation frameworks, leaderboards, and accuracy-efficiency benchmark studies. At present, the methodological contribution appears limited.

8) The manuscript should avoid presenting this narrow empirical study as broad evidence for how open reasoning model evaluation should be conducted in general. The conclusions should be restricted to the tested models, datasets, prompts, hardware, and pipeline.

---

> ### Author Response · Authors · 2026-07-16
> **Response to Reviewer 8gp8 (Part 1 of 3)**
>
> # Response to Reviewer 8gp8
> We sincerely thank Reviewer 8gp8 for the detailed and constructive review. We have uploaded a revised manuscript in which all changes are marked with yellow highlighted text. The revision adds a 500-example larger-sample robustness matrix with a 1000-draw repeated-subset stability analysis, parser-aware failure diagnostics that separate extraction failures from wrong answers, an ARC-Challenge rationale-allowed prompt-protocol ablation, a direct bf16/load-state audit, expanded reproducibility documentation, a weight-sensitivity analysis, Holm-adjusted statistical comparisons, explicitly narrowed claims, and a new Broader Impact section. During revision, we corrected an answer-ordering issue in the prepared TruthfulQA MC1 records and updated the affected evaluations and analyses throughout the manuscript. A detailed point-by-point response is provided below.
>
> ---
>
> **Comment 1.** "The authors should either evaluate on the full benchmark splits or provide a convincing stability analysis over multiple random subsets. A single 238-example subset with one seed is not enough to support the paper's claims."
>
> **Response.** We carefully addressed this in three ways. First, we added a larger-sample robustness matrix in which ARC-Challenge, GSM8K, and TruthfulQA MC1 are evaluated at 500 examples per condition. Second, we clarified that MATH L1–L3 cannot be expanded without changing the task definition: the configured MATH test split contains exactly 43 Level 1, 90 Level 2, and 105 Level 3 examples, so the 238-example MATH component is the complete level-1–3 test subset, not a sampled fraction. Third, we added a repeated-subset stability analysis over 1000 sampled 238-example subsets drawn from the larger non-MATH matrices (with the complete MATH component fixed), reporting mean, SD, 2.5th/97.5th percentiles, and winner frequencies: Gemma-4-26B-A4B zero-shot is the weighted leader in 926/1000 subsets, Gemma-4-E4B few-shot CoT in 72/1000, and Gemma-4-E4B zero-shot in 2/1000, with mean rank correlation 0.985 (SD 0.004) against the matched-core ranking. The main deployment-aware conclusions therefore persist under larger samples and repeated resampling.
>
> **Manuscript location:** Section 3.1; Section 4.2; Tables 2–4; Appendix E, Table 15.
>
> ---
>
> **Comment 2.** "The Phi-4-Reasoning results should be rerun with a more appropriate extraction protocol, or at least reported with parser failures clearly separated from incorrect answers. Treating extraction failures as ordinary benchmark failures gives a misleading impression of model capability."
>
> **Response.** We now explicitly separate strict benchmark scoring from diagnostic parsing. The strict score is retained in all main tables, while a diagnostic lenient parser (which ignores trace-like spans such as `<think>...</think>` and searches the remaining text for likely answer content) is used only for failure attribution, never as a replacement benchmark score. Phi-4-Reasoning is now reported with strict accuracy, diagnostic lenient accuracy, missing-prediction rate, and dominant diagnostic pattern per condition; on GSM8K, its missing-prediction rates are 0.958–0.983 with near-universal think-tag traces, and a separate ARC-Challenge rationale-allowed ablation further shows, at the aggregate level across four representative configurations, that answer-format compliance materially affects multiple-choice accuracy. We accordingly softened the claim from model incapability to poor scoreability and interface adherence under the standardized extraction interface.
>
> **Manuscript location:** Sections 4.7–4.8; Tables 7–8; Appendix B; Appendix F, Table 21.
>
> ---
>
> **Comment 3.** "The authors should explain how CoT prompting is meaningful when the appended task rule says 'return only one capital letter' and 'do not output any explanation.' They should either redesign the prompts or substantially weaken the claims about prompt sensitivity."
>
> **Response.** We agree that these conditions do not constitute unconstrained rationale-generation evaluations. The revised manuscript therefore describes the original multiple-choice CoT conditions as controlled prompt-family variants. To distinguish rationale allowance from answer-format compliance, we added a 500-example ARC-Challenge prompt-protocol ablation in which models may generate up to four brief reasoning steps but must end with a final `#### <letter>` line. Across four representative configurations, aggregate strict accuracy increases from 0.579 to 0.893 for CoT and from 0.638 to 0.823 for few-shot CoT. We accordingly narrowed the prompt-sensitivity claims to clarify that the observed multiple-choice effects reflect both rationale generation and answer-format compliance. TruthfulQA MC1 prompt sensitivity is now interpreted only through the corrected shuffled-choice audit.
>
> **Manuscript location:** Section 3.3; Sections 4.8–4.9; Table 8 (and Table 9 for corrected TruthfulQA); Appendix A.5.
>
> ---
> Continued in Part 2 of 3.

---

> ### Author Response · Authors · 2026-07-16
> **Response to Reviewer 8gp8 (Part 2 of 3)**
>
> **Comment 4.** "The paper should specify exact model revisions, tokenizer versions, selected example IDs, prompt files, extraction rules, raw outputs or representative failures, precision modes, quantization behavior, and software versions. In a deployment-aware paper, ambiguity about bf16 versus 4-bit execution is a serious issue."
>
> **Response.** The revision documents the exact Hugging Face model identifiers with all retained revision evidence (recoverable snapshot/ref hashes where available), dataset sources and preparation rules, selected example identifiers archived with the artifact, exact prompt wrappers and few-shot blocks, strict and diagnostic extraction procedures, and separate software snapshots for the retained main-run environment and the follow-up audit environment. Raw or compact output summaries are archived with the artifact, and representative failure patterns and sampled outputs are discussed in Section 4.7. To resolve the bf16-versus-4-bit ambiguity directly, we performed a load-only audit of all seven model configurations under the same identifiers and loading path: every model reports `is_loaded_in_4bit=false`, `is_loaded_in_8bit=false`, `quantization_config=None`, zero `Linear4bit` modules, zero bitsandbytes modules, and bfloat16 parameter tensors, with load-audit memory footprints consistent with the observed per-model VRAM values. The retained load-mode fields, observed VRAM values, and follow-up direct load audit support interpreting the reported runs as H100, batch-size-one, bf16 executions under the reported stack.
>
> **Manuscript location:** Section 3.4; Section 4.7; Appendices A–C; Tables 11–13; Appendix F, Table 23.
>
> ---
>
> **Comment 5.** "The weighted score should either be justified through a deployment utility model or treated as only one illustrative summary. The paper should avoid making the weighted ranking the central evidence unless the weights are meaningfully motivated."
>
> **Response.** Here, we adopted the second option: the weighted score is now explicitly framed as an illustrative deployment-oriented summary rather than a universal utility function, and the central evidence is the operating-point analysis (Pareto frontier, deployment-budget summaries, and efficiency metrics) rather than the weighted ranking alone. We added a weight-sensitivity analysis over the stated, equal, and four task-heavy weightings for both the matched-core and larger-sample matrices: Gemma-4-26B-A4B zero-shot remains the winner under the stated, ARC-heavy, GSM8K-heavy, and MATH-heavy schemes, while Gemma-4-E4B few-shot CoT wins under equal and TruthfulQA-heavy weighting, which the text now uses to caution against treating any single aggregate vector as decisive.
>
> **Manuscript location:** Section 3.5; Sections 4.3 and 4.5; Appendix F, Table 22.
>
> ---
>
> **Comment 6.** "The authors should explain the bootstrap and permutation procedures more carefully, including the resampling unit and dependence structure. They should also address multiple comparisons and show robustness across sampled subsets."
>
> **Response.** The revision specifies the full procedure: bootstrap intervals use 5000 resamples with examples resampled at the dataset level before recomputing the task-weighted aggregate; paired permutation tests use 20000 sign-flip iterations over aligned example-level correctness differences, preserving the paired structure induced by evaluating the same selected examples across configurations; the global seed is 42. Multiple comparisons are handled with Holm correction over the family of seven reported pairwise comparisons — after correction, several top-row gaps (e.g., leader versus Gemma-4-E4B few-shot CoT: raw p = 0.2132, Holm p = 0.8528) are no longer treated as decisive, and the prose was revised accordingly. Robustness across sampled subsets is shown by the 1000-draw repeated-subset analysis, in which each draw uses the same sampled indices for every configuration within a dataset and the corrected TruthfulQA permutation is fixed across all models and prompts.
>
> **Manuscript location:** Section 3.5; Sections 4.2–4.3; Table 4; Appendix F, Tables 17–18.
>
> ---
> Continued in Part 3 of 3.

---

> ### Author Response · Authors · 2026-07-16
> **Response to Reviewer 8gp8 (Part 3 of 3)**
>
> **Comment 7.** "The paper should make clearer what is new relative to existing LLM evaluation frameworks, leaderboards, and accuracy-efficiency benchmark studies. At present, the methodological contribution appears limited."
>
> **Response.** We rewrote the positioning to state that the contribution is not a larger leaderboard but a deployment-aware evaluation pattern: a complete model–dataset–prompt matrix, shared prompt construction and extraction rules, paired resource measurements, prompt-sensitivity analysis, parser-compliance diagnostics, and operating-point interpretation applied jointly. The revised paper explains what this pattern exposes that accuracy-only leaderboard reporting hides — divergence among score leadership, budget-constrained leadership, and efficiency leadership; benchmark-preparation and answer-ordering artifacts, as illustrated by the corrected TruthfulQA evaluation; and interface-adherence failures — and states that the broader claim is methodological rather than a universal ranking.
>
> **Manuscript location:** Section 2.4; Section 5.
>
> ---
>
> **Comment 8.** "The manuscript should avoid presenting this narrow empirical study as broad evidence for how open reasoning model evaluation should be conducted in general. The conclusions should be restricted to the tested models, datasets, prompts, hardware, and pipeline."
>
> **Response.** We narrowed the claims carefully throughout. The Discussion now opens by stating that the results are operating-point findings for the evaluated model set, benchmark suite, prompt family, hardware environment, precision mode, and extraction pipeline, and that the exact ranking should not be assumed to transfer to other releases, hardware platforms, batch sizes, precision modes, prompts, or extraction systems; the Abstract, Limitations, and Conclusion were scoped in the same way. Regarding the single hardware and batch-size-one configuration specifically: the released artifacts and the load-state audit protocol allow the deployment measurements (latency, VRAM, and load state) to be reproduced and extended on other hardware and software environments, which we identify as future work rather than claiming hardware-invariant values from the present H100 batch-size-one measurements.
>
> **Manuscript location:** Abstract; Section 2.4; Section 3.4; Section 5 (Discussion opening, Limitations, and Future Work); Section 6 (Conclusion).
>
> ---
>
> **Broader Impact Concern.** "I do not see major ethical concerns that would by themselves require rejection. However, I do think the paper should probably include a short broader impact discussion."
>
> **Response.** We added a Broader Impact section. It states that the work aims to improve transparency in open-model evaluation by encouraging reporting of resource use, prompt sensitivity, parser compliance, and output-format robustness alongside accuracy, and it warns that benchmark results can be misused if treated as universal rankings; the reported rankings are context-specific operating points rather than general claims about model intelligence or safety.
>
> **Manuscript location:** Broader Impact section (Section 5, following Limitations).

---

### Author Response · Authors · 2026-07-03
**Acknowledgement**

We thank the reviewers for their careful and constructive feedback. We appreciate the common concerns raised about the matched 238-example design, precision/quantization reporting, extraction robustness, prompting protocol, weighted-score interpretation, and reproducibility details.

We are currently preparing a targeted revision and additional analyses to address these points directly. In particular, we plan to report a sample-size/rank-stability analysis, clarify the actual precision or quantization behavior for each model run, separate parser/extraction failures from incorrect answers, examine stricter versus more lenient extraction where appropriate, clarify the role of CoT-style prompts under answer-format constraints, and expand the reproducibility documentation.

We will follow up with a revised manuscript and detailed point-by-point responses to each reviewer comment as soon as these analyses are complete.

---

### Author Response · Authors · 2026-07-16
**Revised Manuscript and Responses Posted**

We have uploaded the revised manuscript, with all changes highlighted in yellow, and posted detailed point-by-point responses under each reviewer’s review. We thank the reviewers and Action Editor for their feedback.